# MULTILAYER CORRELATION CLUSTERING

## ABSTRACT

We establish Multilayer Correlation Clustering, a novel generalization of Correlation Clustering to the multilayer setting. In this model, we are given a series of inputs of Correlation Clustering (called layers) over the common set $V$ of $n$ elements. The goal is to find a clustering of $V$ that minimizes the $\ell_p$-norm ($p \geq 1$) of the multilayer-disagreements vector, which is defined as the vector (with dimension equal to the number of layers), each element of which represents the disagreements of the clustering on the corresponding layer. For this generalization, we first design an $O(L \log n)$-approximation algorithm, where $L$ is the number of layers. We then study an important special case of our problem, namely the problem with the so-called probability constraint. For this case, we first give an $(\alpha + 2)$-approximation algorithm, where $\alpha$ is any possible approximation ratio for the single-layer counterpart. Furthermore, we design a 4-approximation algorithm, which improves the above approximation ratio of $\alpha + 2 = 4.5$ for the general probability-constraint case. Computational experiments using real-world datasets support our theoretical findings and demonstrate the practical effectiveness of our proposed algorithms.

## 1 INTRODUCTION

Clustering objects based on the information of their similarity is a fundamental task in machine learning. *Correlation Clustering*, introduced by Bansal et al. (2002; 2004), is an optimization model that mathematically formulates this task. In the model, we are given a set $V$ of $n$ elements, where each pair of elements is labeled either '+' (representing that they are similar) or '−' (representing that they are dissimilar) together with a nonnegative weight representing the degree of similarity/dissimilarity. In general, the goal of Correlation Clustering is to find a clustering of $V$ that is consistent with the given similarity information as much as possible. The (in)consistency of a clustering of $V$ can be measured by the so-called *disagreements*, which is defined as the sum of weights of misclassified pairs, i.e., pairs with '+' label across clusters and pairs with '−' label within the same cluster. The problem of finding a clustering of $V$ that minimizes the disagreements is called MINDISAGREE.

It is known that MINDISAGREE is not only NP-hard (Bansal et al., 2002) but also APX-hard even if we consider the unweighted case (i.e., the case where the weights are all equal to 1) (Charikar et al., 2005). A large body of work has been devoted to designing polynomial-time approximation algorithms for the problem. For the general weighted case, Charikar et al. (2005) and Demaine et al. (2006) independently proposed $O(\log n)$-approximation algorithms, using the well-known *region growing* technique (Garg et al., 1996). The approximation ratio of $O(\log n)$ is still the state-of-the-art, and it is also known that improving it is at least as hard as improving the $O(\log n)$-approximation for Minimum Multicut (Garg et al., 1996), which is one of the major open problems in theoretical computer science. For the unweighted case, Bansal et al. (2002) presented the first constant-factor approximation algorithm, which has been improved by a series of works so far (Ailon et al., 2008; Cao et al., 2024; Charikar et al., 2005; Chawla et al., 2015; Cohen-Addad et al., 2022; 2023). Notably, the current-best approximation ratio for the unweighted case is $1.437 + \epsilon$ for any $\epsilon > 0$ (Cao et al., 2024). For more details, see Section 2.

### 1.1 OUR CONTRIBUTION

In this study, we establish *Multilayer Correlation Clustering*, a novel generalization of Correlation Clustering to the multilayer setting. In the model, we are given a series of inputs of Correlation

Clustering (called *layers*) over the common set $V$ of $n$ elements. The goal is then to find a clustering of $V$ that is consistent as much as possible with *all* layers. To quantify the (in)consistency of a clustering over layers, we introduce the concept of *multilayer-disagreements vector* (with dimension equal to the number of layers) of a clustering, each element of which represents the disagreements of the clustering on the corresponding layer.[1] Using the $\ell_p$-norm ($p \geq 1$) of this vector, we can quantify the (in)consistency of the given clustering in a variety of regimes. In particular, if we set $p = 1$, it simply quantifies the sum of disagreements over all layers, whereas if we set $p = \infty$, it quantifies the maximal disagreements over the layers. For $p \geq 1$, our problem asks to find a clustering of $V$ that minimizes the $\ell_p$-norm of the multilayer-disagreements vector.

Multilayer Correlation Clustering is motivated by real-world scenarios. Suppose that we want to find a clustering of users of $\mathbb{X}$ using their similarity information. In this case, various types of similarity can be defined through analysis of users' tweets and observations of different types of connections among users such as follower relations, retweets, and mentions. In the original Correlation Clustering, we need to deal with that information one by one and manage to aggregate resulting clusterings. On the other hand, Multilayer Correlation Clustering enables us to handle that information simultaneously, directly producing a clustering that is consistent (as much as possible) with all types of information. As another example scenario, suppose that we analyze brain networks, where nodes correspond to small regions of a brain and edges represent similarity relations among them. Then it is often the case that the edge set is not determined uniquely; indeed, there would be at least two types of similarity based on the structural connectivity and the functional connectivity among the small pieces of a brain. Obviously, Multilayer Correlation Clustering can again find its advantage in this context.

For this novel, well-motivated generalization, we present a variety of algorithmic results. We first design a polynomial-time $O(L \log n)$-approximation algorithm, where $L$ is the number of layers. Our algorithm is a generalization of the $O(\log n)$-approximation algorithms for MINDISAGREE (Charikar et al., 2005; Demaine et al., 2006) and thus employs the region growing technique (Garg et al., 1996). Our algorithm first solves a convex programming relaxation of the problem. Then, the algorithm iteratively constructs a cluster (and removes it from $V$ as a part of the output), using the region growing technique based on the pseudometric computed by the relaxation, until all elements are clustered. Specifically, in each iteration, the algorithm takes an arbitrary element in $V$ and constructs a ball of center being that element and a radius carefully selected using the similarity information over all layers.

We then study an important special case of our problem, namely the problem with the *probability constraint*, where on each layer, each pair of elements in $V$ has *both* '+' and '−' labels, each of which is associated with a nonnegative weight in $[0, 1]$ and the sum of those two weights is equal to 1. For this problem, we first give a polynomial-time $(\alpha + 2)$-approximation algorithm, where $\alpha$ is any possible approximation ratio for MINDISAGREE with the probability constraint or any of its special cases if we consider the corresponding special case of our problem. For instance, we can take $\alpha = 2.5$ in general (Ailon et al., 2008), $\alpha = 1.437 + \epsilon$ for the unweighted case (Cao et al., 2024), and $\alpha = 1.5$ for the case where the weights of '−' labels satisfy the triangle inequality constraint (see Section 3) (Chawla et al., 2015). In the algorithm design, we first reduce our problem to a novel optimization problem in a metric space, and devise an algorithm to solve it. We then design a 4-approximation algorithm for the general probability-constraint case, improving the above approximation ratio of $\alpha + 2 = 4.5$. The algorithm first solves a convex programming relaxation as in the aforementioned $O(L \log n)$-approximation algorithm, and then constructs a clustering, using a simple thresholding rule. Our algorithm is a generalization of the 4-approximation algorithm for MINDISAGREE of the unweighted case, designed by Charikar et al. (2005).

Finally we conduct thorough experiments using a variety of real-world datasets to evaluate the performance of our proposed algorithms in terms of both solution quality and running time. We confirm that our algorithms outperform baseline methods for both Problem 1 of the general weighted case and Problem 1 with the probability constraint. In particular, the objective value achieved by our algorithm for Problem 1 of the general weighted case is often quite close to the optimal value of the convex programming relaxation, i.e., a lower bound on the optimal value of the problem, meaning that the algorithm tends to obtain a near-optimal solution.

---

[1] It is worth remarking that there is an existing concept called *disagreements vector* in the literature of Correlation Clustering with *fairness* consideration (Kalhan et al., 2019). However, our multilayer-disagreements vector is a different concept from it. For details, see Section 2 and Appendix A.1.

Due to space limitations, we have deferred all proofs of theorems to the Appendix; however, we provide proof ideas and sketches in the main paper.

## 2 RELATED WORK

In this section, we review related literature about special cases and generalizations of MINDISAGREE and multilayer-network analysis.

**Special cases of MINDISAGREE.** For MINDISAGREE of the unweighted case, Bansal et al. (2002; 2004) gave the first constant-factor approximation algorithm with the approximation ratio of 17,429. Then the approximation ratio has been improved by a series of works. Charikar et al. (2005) designed a 4-approximation algorithm. Ailon et al. (2008) then gave KWIKCLUSTER, a purely-combinatorial randomized 3-approximation algorithm. The authors also proved that a variant based on an LP relaxation improves the approximation ratio from 3 to 2.5. Later Chawla et al. (2015) demonstrated that a more sophisticated randomized construction of the clusters achieves a 2.06-approximation (Chawla et al., 2015), which almost matches the integrality gap 2 of the LP relaxation (Charikar et al., 2005). In a recent breakthrough, Cohen-Addad et al. (2022) designed a $(1.994 + \epsilon)$-approximation algorithm for any $\epsilon > 0$, using a semidefinite programming relaxation of the problem, which was further improved to $1.73 + \epsilon$ by introducing a novel preprocessing algorithm (Cohen-Addad et al., 2023). Very recently, Cao et al. (2024) designed a $(1.437 + \epsilon)$-approximation algorithm that runs in $O(n^{\mathrm{poly}(1/\epsilon)})$ time, by inventing a stronger LP called the cluster LP.

For MINDISAGREE with the probability constraint, Bansal et al. (2002; 2004) provided an approximation-preserving reduction from the problem to MINDISAGREE of the unweighted case. Specifically, the authors proved that any $\alpha$-approximation algorithm for MINDISAGREE of the unweighted case yields a $(2\alpha + 1)$-approximation algorithm for MINDISAGREE with the probability constraint. Ailon et al. (2008) demonstrated that the counterparts of KWIKCLUSTER and that combined with the pseudometric computed by the LP relaxation achieve a 5-approximation and a 2.5-approximation, respectively, both of which improved the 9-approximation based on the above reduction with the 4-approximation algorithm for MINDISAGREE of the unweighted case by Charikar et al. (2005). In particular, the approximation ratio of 2.5 is still known to be the state-of-the-art. It is also known that in the case where the weights of '−' labels satisfy the triangle inequality constraint additionally, the approximation ratio can be improved. Indeed, Ailon et al. (2008) proved that their above algorithms achieve a 2-approximation, and later Chawla et al. (2015) improved it to 1.5.

Gionis et al. (2007) studied the problem called *Clustering Aggregation*, which is highly related to MINDISAGREE. In the problem, we are given $L$ clusterings of the common set $V$, and the goal is to find a clustering of $V$ that is consistent with the given clusterings as much as possible. The (in)consistency is measured by the sum of distances between the output clustering and the given $L$ clusterings, where the distance is defined as the number of pairs of elements that are clustered in the opposite way. Gionis et al. (2007) proved that Clustering Aggregation is a special case of MINDISAGREE with the probability constraint and the triangle inequality constraint. We can also directly see that Clustering Aggregation is a quite special case of Multilayer Correlation Clustering of the unweighted case, where each layer already represents a clustering and the parameter $p$ of the $\ell_p$-norm is set to 1. Gionis et al. (2007) also demonstrated that picking up the best clustering among the given $L$ clusterings gives a $2(1 - 1/L)$-approximation while an algorithm similar to the 4-approximation algorithm for MINDISAGREE of the unweighted case (Charikar et al., 2005) achieves a 3-approximation.

**Generalizations of MINDISAGREE.** The most related generalization would be Multi-Chromatic Correlation Clustering (MCCC), introduced by Bonchi et al. (2015), as a further generalization of Chromatic Correlation Clustering (CCC) (Bonchi et al., 2012). Let $V$ be a set of $n$ elements and $C$ a set of colors. Each pair of elements in $V$ is associated with a subset of $C$, meaning that the endpoints are similar in the sense of those colors. The goal is to find a clustering of $V$ and an assignment of each cluster to a subset of $C$ that is consistent as much as possible with the given similarity information. The (in)consistency of a clustering is evaluated as follows: For each pair within a cluster, a distance between the color subsets of the pair and the cluster is charged, while for each pair across clusters, a distance between the color subset of the pair and the emptyset is charged. Varying the definition of the distance, a number of concrete models can be obtained. Although the input of MCCC is essentially

the same as that of our problem of the unweighted case, ours has three concrete advantages: (i) our objective function is more intuitive but can deal with complex relations among the (in)consistency over all layers; (ii) MCCC asks to specify the colors (i.e., layers in our case) of each cluster for which the cluster is supposed to be valid, but our problem does not require such an effort; (iii) our problem is capable of the general weighted case, while MCCC is defined only for the unweighted case and the way to generalize it to the weighted case is not trivial. For MCCC, Bonchi et al. (2015) gave an approximation ratio proportional to the product of $|C|$ and the maximum degree (when interpreting the input as a graph). Recently, Klodt et al. (2021) introduced a different yet similar generalization of CCC to the multi-chromatic case and devised a 3-approximation algorithm based on KWIKCLUSTER.

Multilayer Correlation Clustering can be seen as Correlation Clustering with *fairness* considera-tion (Ahmadi et al., 2019; 2020; Ahmadian et al., 2020; Ahmadian & Negahbani, 2023; Charikar et al., 2017; Davies et al., 2023; Friggstad & Mousavi, 2021; Heidrich et al., 2024; Kalhan et al., 2019; Puleo & Milenkovic, 2016; 2018; Schwartz & Zats, 2022) and *uncertainty* consideration (Chen et al., 2014; Joachims & Hopcroft, 2005; Kuroki et al., 2024; Makarychev et al., 2015; Mathieu & Schudy, 2010). For details, see Appendix A.1.

**Multilayer-network analysis.** Correlation Clustering can be seen as a network clustering model. A *multilayer network* is a generalization of the ordinary network, where we have a number of edge sets (i.e., layers) over the common set of vertices. Multilayer Correlation Clustering can be viewed as a generalization of Correlation Clustering to multilayer networks. Recently, many network-analysis primitives have been generalized from the ordinary networks to multilayer networks. Examples include community detection (Bazzi et al., 2016; De Bacco et al., 2017; Interdonato et al., 2017; Tagarelli et al., 2017), dense subgraph discovery (Galimberti et al., 2020; Jethava & Beerenwinkel, 2015; Kawase et al., 2023), link prediction (De Bacco et al., 2017; Jalili et al., 2017), analyzing spreading processes (De Domenico et al., 2016; Salehi et al., 2015), and identifying central vertices (Basaras et al., 2019; De Domenico et al., 2015).

## 3 PROBLEM FORMULATION

In this section, we formally introduce our problem. Let $V$ be a set of $n$ elements. Let $E$ be the set of unordered pairs of distinct elements in $V$, i.e., $E = \{\{u, v\} : u, v \in V, u \neq v\}$. Let $L$ be a positive integer, representing the number of layers. For each $\ell \in [L]$, let $w_\ell^+ : E \to \mathbb{R}_{\geq 0}$ and $w_\ell^- : E \to \mathbb{R}_{\geq 0}$ be the weight functions for '+' and '−' labels, respectively, on that layer. Note that to deal with the probability constraint case in a unified manner, we assume that each pair of elements has *both* '+' and '−' labels. For simplicity, we define $w_\ell^+(u, v) = w_\ell^+(\{u, v\})$ and $w_\ell^-(u, v) = w_\ell^-(\{u, v\})$ for $\ell \in [L]$ and $\{u, v\} \in E$. Let $\mathcal{C}$ be a clustering (i.e., a partition) of $V$, that is, $\mathcal{C} = \{C_1, \ldots, C_t\}$ such that $\bigcup_{i \in [t]} C_i = V$ and $C_i \cap C_j = \emptyset$ for $i, j \in [t]$ with $i \neq j$. For $v \in V$, we denote by $\mathcal{C}(v)$ the (unique) element (i.e., cluster) in $\mathcal{C}$ to which $v$ belongs. Then, for $u, v \in V$, $\mathbb{1}[\mathcal{C}(u) = \mathcal{C}(v)] = 1$ if $u, v$ belong to the same cluster and $\mathbb{1}[\mathcal{C}(u) \neq \mathcal{C}(v)] = 0$ otherwise. The *disagreement* of $\mathcal{C}$ on layer $\ell \in [L]$ is defined as the sum of weights of misclassified labels on that layer, i.e.,

$$\mathsf{Disagree}_\ell(\mathcal{C}) = \sum_{\{u,v\} \in E} \left( w_\ell^+(u, v) \mathbb{1}[\mathcal{C}(u) \neq \mathcal{C}(v)] + w_\ell^-(u, v) \mathbb{1}[\mathcal{C}(u) = \mathcal{C}(v)] \right).$$

Then the *multilayer-disagreements vector* of $\mathcal{C}$ is defined as $\mathbf{Disagree}(\mathcal{C}) = (\mathsf{Disagree}_\ell(\mathcal{C}))_{\ell \in [L]}$.

We are now ready to formulate our problem:

**Problem 1** (Multilayer Correlation Clustering). *Fix $p \in [1, \infty]$. Given $V$ and $(w_\ell^+, w_\ell^-)_{\ell \in [L]}$, we are asked to find a clustering $\mathcal{C}$ of $V$ that minimizes $\|\mathbf{Disagree}(\mathcal{C})\|_p$, i.e., $\left( \sum_{\ell \in [L]} \mathsf{Disagree}_\ell(\mathcal{C})^p \right)^{1/p}$ if $p < \infty$ and $\max_{\ell \in [L]} \mathsf{Disagree}_\ell(\mathcal{C})$ if $p = \infty$.*

Obviously Problem 1 is a generalization of MINDISAGREE to the multilayer setting. Varying the value of $p$, we can obtain a series of objective functions that evaluate the (in)consistency of the given clustering over the layers in a variety of regimes. If we set $p = 1$, the problem just aims to minimize the sum of disagreements over all layers. It is easy to see that this case can be reduced to MINDISAGREE in an approximation-preserving manner; therefore, the problem is $O(\log n)$-approximable (Charikar et al., 2005; Demaine et al., 2006). If we set $p = \infty$, the problem aims

to minimize the maximal disagreements over all layers, which is an important special case we are particularly interested in.

An important special case of Problem 1 is that $w_\ell^+, w_\ell^-$ for every layer $\ell \in [L]$ satisfy the so-called *probability constraint*, i.e., $w_\ell^+(u,v) + w_\ell^-(u,v) = 1$ for any $\{u,v\} \in E$. Note that the most fundamental special case, i.e., the unweighted case, is still contained in this case, where $w_\ell^-(u,v) = 1 - w_\ell^+(u,v) = 0$ or $1$. Another special case, which we also handle in the present paper, is Problem 1 with the probability constraint and the *triangle inequality constraint*. The triangle inequality constraint stipulates that on every layer $\ell \in [L]$, $w_\ell^-(u,w) \leq w_\ell^-(u,v) + w_\ell^-(v,w)$ holds for any distinct $u,v,w \in V$. It is easy to see that in the case of $p = 1$, Problem 1 with the probability constraint (and the triangle inequality constraint) can be reduced to MINDISAGREE with the probability constraint (and the triangle inequality constraint) in an approximation-preserving manner. Indeed, simply summing up the weights over all layers for each pair of elements and dividing it by $L$, we can obtain an equivalent instance of MINDISAGREE with the probability constraint (and the triangle inequality constraint). Therefore, we see that the problem is still 2.5-approximable (Ailon et al., 2008) in the probability constraint case and 1.5-approximable (Chawla et al., 2015) in the probability constraint and triangle inequality constraint case. Note however that for Problem 1 of the unweighted case, there is no trivial reduction that can beat the above 2.5-approximation.

## 4 ALGORITHM FOR PROBLEM 1

In this section, we design an $O(L \log n)$-approximation algorithm for Problem 1.

### 4.1 THE PROPOSED ALGORITHM

We first present 0–1 convex programming formulations for Problem 1. For distinct $i,j \in V$, we introduce 0–1 variables $x_{ij}, x_{ji}$, both of which take 0 if $i,j$ belong to the same cluster and 1 otherwise. Then, in the case of $p < \infty$, Problem 1 can be formulated as follows:

$$\text{minimize} \quad \left( \sum_{\ell \in [L]} \left( \sum_{\{i,j\} \in E} \left( w_\ell^+(i,j)x_{ij} + w_\ell^-(i,j)(1 - x_{ij}) \right) \right)^p \right)^{1/p}$$

$$\text{subject to} \quad x_{ij} = x_{ji} \quad (\forall i,j \in V,\ i \neq j), \tag{1}$$

$$x_{ik} \leq x_{ij} + x_{jk} \quad (\forall i,j,k \in V,\ i \neq j,\ j \neq k,\ k \neq i), \tag{2}$$

$$x_{ij} \in \{0,1\} \quad (\forall i,j \in V,\ i \neq j). \tag{3}$$

On the other hand, in the case of $p = \infty$, we have the following 0–1 LP formulation:

$$\text{minimize} \quad t$$

$$\text{subject to} \quad \sum_{\{i,j\} \in E} \left( w_\ell^+(i,j)x_{ij} + w_\ell^-(i,j)(1 - x_{ij}) \right) \leq t \quad (\forall \ell \in [L]),$$

$$\text{Constraints (1)--(3).}$$

For the above formulations, by relaxing the constraints $x_{ij} \in \{0,1\}$ to $x_{ij} \in [0,1]$ for all distinct $i,j \in V$, we can obtain continuous relaxations of Problem 1, which we refer to as (CV) and (LP), respectively. Let $\boldsymbol{x} = (x_{ij})_{i,j \in V: i \neq j}$. It should be noted that (CV) is a convex programming problem. Indeed, the objective function is convex, as it is a vector composition of form $f(g(\boldsymbol{x})) = f(g_1(\boldsymbol{x}), \ldots, g_L(\boldsymbol{x}))$, where $f: \mathbb{R}_{\geq 0}^L \to \mathbb{R}_{\geq 0}$ is an $\ell_p$-norm of $p \geq 1$, which is convex and non-decreasing in each argument, and $g_\ell: \mathbb{R}_{\geq 0}^E \to \mathbb{R}_{\geq 0}$ is linear and thus convex for every $\ell \in [L]$; moreover, the set of feasible solutions is obviously convex. Therefore, we can solve the problem to arbitrary precision in polynomial time, using an appropriate method for convex programming such as an interior-point method (Boyd & Vandenberghe, 2004). For simplicity, we suppose that (CV) can be solved exactly in polynomial time. On the other hand, (LP) is indeed an LP, and thus can be solved exactly in polynomial time. Let $\text{OPT}_{\text{CV}}$ and $\text{OPT}_{\text{LP}}$ be the optimal values of the above relaxations, respectively.

Our algorithm first solves an appropriate relaxation, (CV) or (LP), depending on the value of $p$, and obtains its optimal solution $\boldsymbol{x}^* = (x_{ij}^*)_{i,j \in V: i \neq j}$. Then the algorithm updates $\boldsymbol{x}^*$ so that

---

**Algorithm 1:** $O(L \log n)$-approximation algorithm for Problem 1

---

**Input:** $V$ and $(w_\ell^+, w_\ell^-)_{\ell \in [L]}$    **Output:** Clustering of $V$

**1** Compute an optimal solution $\boldsymbol{x}^* = (x_{ij}^*)_{i,j \in V : i \neq j}$ to (CV) if $p < \infty$ and (LP) if $p = \infty$;

**2** Update $\boldsymbol{x}^*$ so that $\boldsymbol{x}^* = (x_{ij}^*)_{i,j \in V}$ by setting $x_{ii}^* = 0$ for every $i \in V$;

**3** Take an arbitrary $c > 2$;

**4** $\mathcal{B} \leftarrow \emptyset$, $V^{(1)} \leftarrow V$, and $t \leftarrow 1$;

**5** **while** $V^{(t)} \neq \emptyset$ **do**

**6**    Take an arbitrary pivot $i^{(t)} \in V^{(t)}$;

**7**    Compute $r_{(t)}^* \in \operatorname{argmin} \left\{ \max\limits_{\ell \in [L]:\, F_\ell \neq 0} \dfrac{\operatorname{cut}_{(V^{(t)},\ell)}(B_{V^{(t)}}(i^{(t)}, r))}{\operatorname{vol}_{(V^{(t)},\ell)}(B_{V^{(t)}}(i^{(t)}, r))} \,:\, r \in (0, 1/c) \right\}$;

**8**    $\mathcal{B} \leftarrow \mathcal{B} \cup \{B_{V^{(t)}}(i^{(t)}, r_{(t)}^*)\}$, $V^{(t+1)} \leftarrow V^{(t)} \setminus B_{V^{(t)}}(i^{(t)}, r_{(t)}^*)$, and $t \leftarrow t + 1$;

**9** **return** $\mathcal{B}$;

---

$\boldsymbol{x}^* = (x_{ij}^*)_{i,j \in V}$ by setting $x_{ii}^* = 0$ for every $i \in V$. Obviously $\boldsymbol{x}^*$ is a pseudometric over $V$, i.e., a relaxed metric where a distance between distinct elements may be equal to 0. Based on this, the algorithm constructs a clustering in an iterative manner: The algorithm initially has the entire set $V$. In each iteration, the algorithm takes an arbitrary element called a *pivot* in the current set and constructs a cluster by collecting the pivot itself and the other elements that are located at distance less than some carefully-chosen value from the pivot. The algorithm removes the cluster from the current set and repeats the process until it is left with the emptyset.

To describe the algorithm formally, we introduce notation. Without loss of generality, we can assume that at most one of $w_\ell^+(u, v)$ and $w_\ell^-(u, v)$ is nonzero for any $\ell \in [L]$ and $\{u, v\} \in E$. Otherwise we can transform the instance into another one that satisfies the above and is more easily approximable (see Section 1.4 in Bonchi et al. (2022) for details). Based on the assumption, for each $\ell \in [L]$, we introduce two mutually-disjoint sets $E_\ell^+ = \{\{u, v\} \in E : w_\ell^+(u, v) > 0\}$ and $E_\ell^- = \{\{u, v\} \in E : w_\ell^-(u, v) > 0\}$, and define $w_\ell \colon E_\ell^+ \cup E_\ell^- \to \mathbb{R}_{>0}$ such that $w_\ell(u, v) = w_\ell^+(u, v)$ if $\{u, v\} \in E_\ell^+$ and $w_\ell(u, v) = w_\ell^-(u, v)$ if $\{u, v\} \in E_\ell^-$. Let $U$ be an arbitrary subset of $V$. For $i \in U$ and $r \geq 0$, we denote by $B_U(i, r)$ the open ball of center $i$ and radius $r$ in $U$, i.e., $B_U(i, r) = \{j \in U : x_{ij}^* < r\}$. For $B_U(i, r)$, we define its cut value $\operatorname{cut}_{(U,\ell)}(B_U(i, r))$ within $U$ on layer $\ell \in [L]$ as the sum of weights of '+' labels across $B_U(i, r)$ and $U \setminus B_U(i, r)$ on layer $\ell \in [L]$, i.e.,

$$\operatorname{cut}_{(U,\ell)}(B_U(i, r)) = \sum_{\{j,k\} \in E_\ell^+ :\, j \in B_U(i,r) \wedge k \in U \setminus B_U(i,r)} w_\ell(j, k).$$

For $B_U(i, r)$, we define its volume $\operatorname{vol}_{(U,\ell)}(B_U(i, r))$ within $U$ on layer $\ell \in [L]$ as

$$\operatorname{vol}_{(U,\ell)}(B_U(i, r)) = \frac{F_\ell}{n} + \sum_{\{j,k\} \in E_\ell^+ :\, j,k \in B_U(i,r)} w_\ell(j, k) x_{jk}^* + \sum_{\{j,k\} \in E_\ell^+ :\, j \in B_U(i,r) \wedge k \in U \setminus B_U(i,r)} w_\ell(j, k)(r - x_{ij}^*),$$

where $F_\ell = \sum_{\{j,k\} \in E_\ell^+} w_\ell(j, k) x_{jk}^*$.

Our formal algorithm is presented in Algorithm 1. The feature can be found in the radius selection: In the $t$-th iteration, the algorithm selects the radius $r_{(t)}^*$ that minimizes the maximal ratio of the cut value to the volume of the ball of the chosen pivot $i^{(t)}$ over all layers $\ell \in [L]$ with $F_\ell \neq 0$. Here we give an intuitive explanation of the role of the volume. If the radius just minimized the cut value, then the cluster would tend to be quite small; consequently, the resulting clustering would consist of a lot of small clusters, which overall causes large disagreements for the pairs of elements with '+' labels. The volume helps avoid this situation. Indeed, thanks to it, the algorithm tends to *consume* a relatively large part of the remaining set, resulting in relatively large clusters.

## 4.2 ANALYSIS OF ALGORITHM 1

We have the following key lemma:

**Lemma 1.** *In Algorithm 1, for any $t = 1, \ldots, |\mathcal{B}|$, it holds that*

$$\max_{\ell \in [L]: F_\ell \neq 0} \frac{\operatorname{cut}_{(V^{(t)}, \ell)}(B_{V^{(t)}}(i^{(t)}, r^*_{(t)}))}{\operatorname{vol}_{(V^{(t)}, \ell)}(B_{V^{(t)}}(i^{(t)}, r^*_{(t)}))} \leq cL \log(n + 1),$$

*and moreover, $B_{V^{(t)}}(i^{(t)}, r^*_{(t)})$ can be computed in $O(Ln^2)$ time.*

Let $\mathcal{B}$ be the output of Algorithm 1. Our analysis is layer-wise, but it directly leads to the evaluation of the disagreements over layers. The disagreements of $\mathcal{B}$ produced by the pairs of elements with '+' labels on layer $\ell \in [L]$ with $F_\ell \neq 0$ equal the sum of weights of '+' labels for those pairs across clusters in $\mathcal{B}$, which can be upper bound by $O(L \log n)$ times the sum of volumes of clusters in $\mathcal{B}$, using Lemma 1. As the sum of volumes is further upper bounded by the sum of corresponding terms in the optimal objective to (CV) or (LP), we can obtain an $O(L \log n)$-approximation for that part. The disagreements of $\mathcal{B}$ produced by the other pairs are easily upper bounded. We have the following theorem:

**Theorem 1.** *Algorithm 1 is a polynomial-time $O(L \log n)$-approximation algorithm for Problem 1. Specifically, the time complexity is $O(T_{\mathrm{CV}} + Ln^3)$ if $p < \infty$ and $O(T_{\mathrm{LP}} + Ln^3)$ if $p = \infty$, where $T_{\mathrm{CV}}$ and $T_{\mathrm{LP}}$ denote the time complexities required to solve (CV) and (LP), respectively.*

Finally we mention the integrality gaps of (CV) and (LP). For MINDISAGREE, the LP relaxation used in the $O(\log n)$-approximation algorithms is known to have the integrality gap of $\Omega(\log n)$ (Charikar et al., 2005; Demaine et al., 2006). As our relaxations, (CV) and (LP), are its generalizations, the integrality gap of $\Omega(\log n)$ is inherited. This matches our approximation ratio in the case of $L = O(1)$ but there remains a gap in general.

## 5 ALGORITHMS FOR PROBLEM 1 WITH PROBABILITY CONSTRAINT

In this section, we present our algorithms for Problem 1 with the probability constraint. The first algorithm has an approximation ratio of $\alpha + 2$, where $\alpha$ is any possible approximation ratio for MINDISAGREE with the probability constraint or any of its special cases if we consider the corresponding special case of our problem. The second algorithm has an approximation ratio of 4.

### 5.1 THE $(\alpha + 2)$-APPROXIMATION ALGORITHM

To design the algorithm, we reduce Problem 1 with the probability constraint to a novel optimization problem in a metric space. Let $X$ be a set. Let $d \colon X \times X \to \mathbb{R}_{\geq 0}$ be a *metric* on $V$, i.e., $d(x, y) = 0$ if and only if $x = y$ for $x, y \in V$, $d(x, y) = d(y, x)$ for $x, y \in V$, and $d(x, z) \leq d(x, y) + d(y, z)$ for $x, y, z \in V$. In general, $(X, d)$ is called a *metric space*. We introduce the following problem:

**Problem 2** (Find the Most Representative Candidate in a Metric Space)**.** *Fix $p \geq 1$. Let $(X, d)$ be a metric space. Given $x_1, \ldots, x_L \in X$ and a candidate set $F \subseteq X$, we are asked to find $x \in F$ that minimizes $\left( \sum_{\ell \in [L]} d(x, x_\ell)^p \right)^{1/p}$ if $p < \infty$ and $\max_{\ell \in [L]} d(x, x_\ell)$ if $p = \infty$.*

Then we can prove the following key lemma. The proof is based on the fact that each layer of the input of Problem 1 with the probability constraint (i.e., an input of MINDISAGREE with the probability constraint) and any clustering of $V$ can be dealt with in a unified metric space $(X, d)$ when $X$ and $d$ are set appropriately.

**Lemma 2.** *There exists a polynomial-time approximation-preserving reduction from Problem 1 with the probability constraint to Problem 2.*

In what follows, we design an approximation algorithm for Problem 2, resulting in an approximation algorithm for Problem 1 with the probability constraint having the same approximation ratio. To this end, we introduce the following subproblem:

**Problem 3** (Find the Closest Candidate in a Metric Space)**.** *Let $(X, d)$ be a metric space. Given $x \in X$ and a candidate set $F \subseteq X$, we are asked to find $x' \in F$ that minimizes $d(x, x')$.*

Assume now that we have an $\alpha$-approximation algorithm for Problem 3. Let $x_1, \ldots, x_L \in X$ and $F \subseteq X$ be the input of Problem 2. Our approximation algorithm for Problem 2 runs as follows: For

---

**Algorithm 2:** $(\alpha + 2)$-approximation algorithm for Problem 2

---

**Input:** $x_1, \ldots, x_L \in X$ and $F \subseteq X$   **Output:** $x \in F$

1 **for** $\ell \in [L]$ **do** $x'_\ell \leftarrow \alpha$-approximate solution for Problem 3 with input $x_\ell \in X$ and $F \subseteq X$;

2 **return** $x_{\text{out}} \in \operatorname{argmin}_{x \in \{x'_1, \ldots, x'_L\}} \left( \sum_{\ell \in [L]} d(x, x_\ell)^p \right)^{1/p}$ *if* $p < \infty$ *and*

$x_{\text{out}} \in \operatorname{argmin}_{x \in \{x'_1, \ldots, x'_L\}} \max_{\ell \in [L]} d(x, x_\ell)$ *if* $p = \infty$;

---

**Algorithm 3:** 4-approximation algorithm for Problem 1 with the probability constraint

---

**Input:** $V$ and $(w^+_\ell, w^-_\ell)_{\ell \in [L]}$   **Output:** Clustering of $V$

1 Perform Lines 1 and 2 in Algorithm 1;

2 Initialize $\mathcal{B} \leftarrow \emptyset$ and $U \leftarrow V$;

3 **while** $U \neq \emptyset$ **do**

4 $\quad$ Take an arbitrary $i \in U$ and initialize $B \leftarrow \{i\}$;

5 $\quad$ $C \leftarrow B_U(i, 1/2) \setminus \{i\}$;

6 $\quad$ **if** $\frac{1}{|C|} \sum_{j \in C} x^*_{ij} < 1/4$ **then** $B \leftarrow B \cup C$;

7 $\quad$ $\mathcal{B} \leftarrow \mathcal{B} \cup \{B\}$ and $U \leftarrow U \setminus B$;

8 **return** $\mathcal{B}$;

---

every $\ell \in [L]$, the algorithm obtains an $\alpha$-approximate solution $x'_\ell \in F$ for Problem 3 with input $x_\ell \in X$ and $F \subseteq X$, using the $\alpha$-approximation algorithm for Problem 3. Then the algorithm outputs the best solution among $x'_1, \ldots, x'_L$ in terms of the objective function of Problem 2. The pseudocode is given in Algorithm 2.

**Analysis.** We analyze the approximation ratio of Algorithm 2. Let $x^* \in F$ be an optimal solution to Problem 2. Let $x_{\text{closest}} \in \operatorname{argmin}_{x \in \{x_1, \ldots, x_L\}} d(x, x^*)$ and $x'_{\text{closest}}$ be the $\alpha$-approximate solution for Problem 3 with input $x_{\text{closest}}$ and $F$. By repeatedly applying the triangle inequality over $d$, we can obtain $d(x'_{\text{closest}}, x_\ell) \leq (\alpha + 2) \cdot d(x^*, x_\ell)$ for any $\ell \in [L]$. Noticing the facts that $x'_{\text{closest}}$ is one of the candidates of the output of the algorithm and that the evaluation of the point-wise distance directly leads to the evaluation of the objective value of Problem 2, we have the following theorem:

**Theorem 2.** *Algorithm 2 is an $(\alpha + 2)$-approximation algorithm for Problem 2.*

In Algorithm 2, the approximation ratio of $\alpha$ for Problem 3 that we can take depends on the metric space $(X, d)$ and part of input $F \subseteq X$, inherited from Problem 2. By interpreting Problem 1 with the probability constraint (or any of its special cases) as Problem 2 with specific metric space $(X, d)$ and part of input $F \subseteq X$, we can obtain the following series of approximability results:

**Corollary 1.** *(i) There exists a polynomial-time $4.5$-approximation algorithm for Problem 1 with the probability constraint. (ii) For any $\epsilon > 0$, there exists a polynomial-time $(3.437 + \epsilon)$-approximation algorithm for Problem 1 of the unweighted case. (iii) There exists a polynomial-time $3.5$-approximation algorithm for Problem 1 with the probability constraint and the triangle inequality constraint.*

## 5.2 The 4-approximation algorithm

Our algorithm first obtains $\boldsymbol{x}^* = (x^*_{ij})_{i,j \in V}$ in exactly the same way as that of Algorithm 1. Based on the pseudometric $\boldsymbol{x}^*$ over $V$, the algorithm then constructs a clustering, using a simple thresholding rule. Let $U$ be an arbitrary subset of $V$. For $i \in U$ and $r \geq 0$, we denote by $B_U(i, r)$ the closed ball of center $i$ and radius $r$ in $U$, i.e., $B_U(i, r) = \{j \in U : x^*_{ij} \leq r\}$. Our algorithm initially set $U = V$. In each iteration, the algorithm takes an arbitrary element $i \in U$ and initializes a cluster $B = \{i\}$. Then the algorithm constructs $C = B_U(i, 1/2) \setminus \{i\}$. If the average distance between $i$ and the elements in $C$ is less than $1/4$, i.e., $\frac{1}{|C|} \sum_{j \in C} x^*_{ij} < 1/4$, then the algorithm updates $B$ by adding all elements in $C$. The algorithm removes $B$ from $U$ as a cluster of the output, and repeats the procedure until $U = \emptyset$. The pseudocode is presented in Algorithm 3.

**Analysis.** The intuition of the analysis is similar to that of Algorithm 1. Based on the thresholding rule together with the probability constraint, we can obtain the approximation ratio of 4:

Table 1: Real-world datasets and experimental results for Problem 1 of the general weighted case.

| Dataset | $|V|$ | $L$ | LB | Algorithm 1 | | Pick-a-Best | | Aggregate | |
|---|---|---|---|---|---|---|---|---|---|
| | | | | Obj. val. | Time(s) | Obj. val. | Time(s) | Obj. val. | Time(s) |
| aves-sparrow-social | 52 | 2 | 13.37 | **13.48** | 0.47 | 26.79 | 0.34 | 13.81 | **0.11** |
| insecta-ant-colony1 | 113 | 41 | 32.48 | **34.30** | 587.94 | 42.94 | 1719.11 | 47.59 | **48.03** |
| reptilia-tortoise-network-bsv | 136 | 4 | 127.14 | **151.00** | 2.32 | 193.00 | 16.43 | 174.00 | **0.91** |
| aves-wildbird-network | 202 | 6 | 54.97 | **56.50** | 35.78 | 98.27 | 129.20 | 74.84 | **7.87** |
| aves-weaver-social | 445 | 23 | 132.75 | **164.00** | 135.22 | — | OT | 177.00 | **12.19** |
| reptilia-tortoise-network-fi | 787 | 9 | 271.48 | **305.00** | 644.07 | — | OT | 446.00 | **195.40** |

**Theorem 3.** *Algorithm 3 is a 4-approximation algorithm for Problem 1 with the probability constraint.*

The above theorem indicates that the 4-approximation algorithm for MINDISAGREE of the unweighted case, designed by Charikar et al. (2005), can be extended to the probability constraint case, which has yet to be mentioned before. Although some approximation ratios better than 4 are known for MINDISAGREE of the unweighted case, thanks to its simplicity and extendability, the algorithm has been generalized to various settings of the unweighted case (see Section 2). Our analysis implies that those results may be further generalized form the unweighted case to the probability constraint case.

## 6 EXPERIMENTAL EVALUATION

In this section, we report the results of computational experiments performed on various real-world datasets, evaluating the practical performance of our proposed algorithms. Due to space limitations, we discuss only Problem 1 of the general weighted case in the main paper. For Problem 1 with the probability constraint, see Appendix D.

### 6.1 EXPERIMENTAL SETUP

**Datasets.** Throughout the experiments, we set $p = \infty$ in Problem 1, meaning that we aim to minimize the maximal disagreements over all layers. This is an important case of particular interest to us, where the objective is quite intuitive and easy to interpret. Table 1 lists real-world datasets, each of which is a multilayer network consisting of $L$ layers with positive edge weights, collected by Network Repository (Rossi & Ahmed, 2015) licensed under a Creative Commons Attribution-ShareAlike License.[2] Using the datasets, we generated our instances of Problem 1. Let $G = (V, (E_\ell, w_\ell)_{\ell \in [L]})$ be a multilayer network at hand, where $E_\ell$ is the set of edges on layer $\ell$ and $w_\ell \colon E_\ell \to \mathbb{R}_{>0}$ is its weight function. We first normalize all edge weights so that the maximum weight over layers is equal to 1; that is, we redefine $w_\ell(\{u,v\}) \leftarrow w_\ell(\{u,v\})/w_{\max}$ for every $\ell \in [L]$ and $\{u,v\} \in E_\ell$, where $w_{\max} = \max_{\ell \in [L]} \max_{\{u,v\} \in E_\ell} w_\ell(\{u,v\})$. For every $\ell \in [L]$, let $\mathtt{weights}(\ell)$ be the multiset of all edge weights on layer $\ell$, i.e., $\mathtt{weights}(\ell) = \{w_\ell(\{u,v\}) : \{u,v\} \in E_\ell\}$. We generate our instance $V$ and $(w_\ell^+, w_\ell^-)_{\ell \in [L]}$ as follows: The set $V$ of objects is exactly the same as the set of vertices in the multilayer network. For convenience, we define $E = \{\{u,v\} : u, v \in V, u \neq v\}$. For each layer $\ell \in [L]$ and $\{u,v\} \in E$, if $\{u,v\} \in E_\ell$ we set $w_\ell^+(u,v) = w_\ell(\{u,v\})$ and $w_\ell^-(u,v) = 0$; otherwise we set $w_\ell^+(u,v) = 0$ and $w_\ell^-(u,v) = \mathrm{Uniform}(\mathtt{weights}(\ell))$ with probability 0.5, where $\mathrm{Uniform}()$ takes an element from a given multiset uniformly at random, and $w_\ell^+(u,v) = w_\ell^-(u,v) = 0$ otherwise. The intuition behind the above setting is that we actively put '+' labels for the pairs of objects having edges, while for the pairs of objects not having edges, we only passively put '−' labels (i.e., only with probability 0.5), given the potential missing of edges in the original network. The weights for '+' labels fully respect for the original edge weights, while weights for '−' labels are generated from those for '+' labels.

**Our algorithms and baselines.** In Algorithm 1, the way to select a pivot is arbitrary; in our implementation, the algorithm just takes the object with the smallest ID. We employ the following two baseline methods: (i) Pick-a-Best: This method first solves MINDISAGREE on each layer, using the state-of-the-art $O(\log n)$-approximation algorithms (Charikar et al., 2005; Demaine et al., 2006), and then outputs the best one among them in terms of the objective value of Problem 1.

---

[2]https://networkrepository.com/index.php

This method can be seen as a generalization of Algorithm 2 for Problem 1 with the probability constraint case, but it is not clear if the method has an approximation ratio such as $O(L \log n)$, achieved by Algorithm 1. (ii) Aggregate: This method first aggregates the layers. Specifically, the method constructs $w^+ \colon E \to \mathbb{R}_{\geq 0}$ and $w^- \colon E \to \mathbb{R}_{\geq 0}$ by setting $w^+(u, v) = \sum_{\ell \in [L]} w_\ell^+(u, v)$ and $w^-(u, v) = \sum_{\ell \in [L]} w_\ell^-(u, v)$ for every $\{u, v\} \in E$. Then it solves MINDISAGREE with input $V$ and $(w^+, w^-)$, using the $O(\log n)$-approximation algorithms (Charikar et al., 2005; Demaine et al., 2006). As mentioned in Section 3, this method gives an $O(\log n)$-approximate solution for Problem 1 when $p = 1$, but the approximation ratio for the case of $p = \infty$ is not clear.

Finally we mention the implementation of the LPs. All LPs here have the $\Theta(n^3)$ triangle inequality constraints; thus, it is inefficient to input the entire program directly. To overcome this, we employed Row Generation technique (Grötschel & Wakabayashi, 1989). Specifically, we first solve the program without any triangle inequality constraint. Then we scan all the constraints: If there are constraints violated by the current optimal solution, we add the constraints to the program, solve it again, and repeat the process; otherwise we output the current optimal solution, which is an optimal solution to the original program.

**Machine spec and code.** We used a machine with Apple M1 Chip and 16 GB RAM. All codes were written in Python 3. LPs were solved using Gurobi Optimizer 11.0.1 with the default parameters.

### 6.2 RESULTS

The results are presented in Table 1, where for each instance, the best objective value and running time among the algorithms are written in bold. The fourth column, named LB, presents $\text{OPT}_{\text{LP}}$, i.e., the optimal value of (LP), which is a lower bound on the optimal value of Problem 1. OT indicates that the algorithm did not terminate in 3,600 seconds. As can be seen, Algorithm 1 outperforms the baseline methods in terms of the quality of solutions. Indeed, Algorithm 1 obtains much better solutions than those computed by Pick-a-Best and Aggregate. Remarkably, the objective value achieved by Algorithm 1 is often quite close to the lower bound $\text{OPT}_{\text{LP}}$, meaning that the algorithm tends to obtain a near-optimal solution. As Algorithm 1 solves (LP), which involves the multilayer structure and thus is more complex than the LP solved in Aggregate, Algorithm 1 is slower than Aggregate; however, Algorithm 1 is still even faster than Pick-a-Best, as the latter requires to solve $L$ different LPs corresponding to the layers.

## 7 CONCLUSIONS

We have introduced Multilayer Correlation Clustering, a novel generalization of Correlation Clustering to the multilayer setting, and designed approximation algorithms. As a final remark, we discuss the limitations of our work, based on which we mention several interesting open problems. In theory, it is still not clear how harder Multilayer Correlation Clustering is to approximate compared with MINDISAGREE. Given this situation, we believe that the most promising direction is to fill the gap: Improve the approximation ratios achieved by our proposed algorithms and/or proving some hardness of approximation for Multilayer Correlation Clustering (beyond that for MINDISAGREE). One of the reasonable questions is "to what extent can we reduce the term $L$ in the current approximation ratio of $O(L \log n)$ of Algorithm 1?" In practice, our algorithms that solve LPs do not scale to large instances. Therefore, it is also interesting to (further) investigate fast algorithms for Multilayer Correlation Clustering even without approximation ratios. For the detailed descriptions of open problems, see Appendix E.1.

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

## A  OMITTED CONTENTS IN SECTION 2

### A.1  DETAILS OF RELATED WORK

Multilayer Correlation Clustering can be seen as Correlation Clustering with *fairness* considerations. Indeed, supposing that the similarity information of each layer is given by an agent (e.g., a crowd worker), we see that the problem tries not to abandon any similarity information given by the agents. From a fairness perspective, Puleo & Milenkovic (2016; 2018) initiated the study of local objectives

for MINDISAGREE of the unweighted case. In this model, the disagreements of a clustering are quantified locally rather than globally, at the level of single elements. Specifically, they considered a disagreements vector (with dimension equal to the number of elements), where $i$-th element represents the disagreements incident to the corresponding element $i \in V$. The goal is then to minimize the $\ell_p$-norm ($p \geq 1$) of the disagreements vector. If we set $p = 1$, the problem reduces to MINDISAGREE of the unweighted case, whereas if we set $p = \infty$, the problem aims to minimize the maximal disagreements over the elements. The authors proved that the model with $p = \infty$ is NP-hard and designed a 48-approximation algorithm for any $p \geq 1$ by extending the 4-approximation algorithm for MINDISAGREE of the unweighted case, designed by Charikar et al. (2005). Charikar et al. (2017) then improved the approximation ratio to 7 by inventing a different rounding algorithm. The contribution of Charikar et al. (2017) is not limited to the unweighted case; they also studied the above model with $p = \infty$ of the general weighted case and designed an $O(\sqrt{n})$-approximation algorithm. Later Kalhan et al. (2019) improved the above approximation ratio of 7 to 5, and designed an $O(n^{\frac{1}{2} - \frac{1}{2p}} \log^{\frac{1}{2} + \frac{1}{2p}} n)$-approximation algorithm for any $p \geq 1$ of the general weighted case, matching the current-best approximation ratio of $O(\log n)$ for MINDISAGREE of the general weighted case (i.e., the above model with $p = 1$) (Charikar et al., 2005; Demaine et al., 2006). Davies et al. (2023) gave a purely-combinatorial $O(n^\omega)$-time 40-approximation algorithm for $p = \infty$ of the unweighted case, where $\omega$ is the exponent of matrix multiplication, while Heidrich et al. (2024) improved the above approximation ratio of 5 by Kalhan et al. (2019) to 4 for $p = \infty$. Very recently, Davies et al. (2024) designed a combinatorial algorithm running in $O(n^\omega)$ time and outputting a clustering that is a constant-factor approximate solution for all $\ell_p$-norms simultaneously. Ahmadi et al. (2019) studied the cluster-wise counterpart of the above model with $p = \infty$ (of the general weighted case), where the goal is to find a clustering of $V$ that minimizes the maximal disagreements over the clusters. The authors presented an $O(\log n)$-approximation algorithm together with an $O(r^2)$-approximation algorithm for the $K_{r,r}$-free graphs. Later Kalhan et al. (2019) significantly improved these approximation ratios to $2 + \epsilon$ for any $\epsilon > 0$.

Another type of fairness has been considered for Correlation Clustering. Ahmadian et al. (2020) initiated the study of Fair Correlation Clustering (of the unweighted case), where each element is associated with a color, and each cluster of the output is required to be not over-represented by any color, meaning that the fraction of elements with any single color has to be upper bounded by a specified value. For the model, the authors designed a 256-approximation algorithm, based on the notion called fairlet decomposition. Ahmadi et al. (2020) independently studied a similar model of Fair Correlation Clustering, where the distribution of colors in each cluster has to be the same as that of the entire set. In particular, for the case of two colors that have the same number of elements in the entire set, the authors proposed a $(3\alpha + 4)$-approximation algorithm, where $\alpha$ is any known approximation ratio for MINDISAGREE of the unweighted case. Friggstad & Mousavi (2021) then gave an approximation ratio of 6.18, which cannot be achieved by the above $3\alpha + 4$. The authors also studied the model with the aforementioned local objective for $p = \infty$ and designed a constant-factor approximation algorithm. Schwartz & Zats (2022) proved that the model of Ahmadi et al. (2020) of the general weighted case has no finite approximation ratio, unless P = NP. Very recently, Ahmadian & Negahbani (2023) substantially generalized the above models and designed an approximation algorithm that has constant-factor approximation ratios for some useful special cases.

Multilayer Correlation Clustering can also be seen as Correlation Clustering with the *uncertainty* of input by interpreting each layer as a possible scenario of the similarity information of the elements. Most works on Correlation Clustering with uncertainty assume the existence of the ground-truth clustering of $V$ and aim to recover it, based only on its noisy observations. In the seminal paper by Bansal et al. (2004), this type of problem had already been considered, while Joachims & Hopcroft (2005) gave the first formal analysis of the problem. Later, a variety of problem settings have been introduced in a series of works (Chen et al., 2014; Makarychev et al., 2015; Mathieu & Schudy, 2010; Silwal et al., 2023). Very recently, Kuroki et al. (2024) considered another type of problem, which aims to perform as few queries as possible to an oracle that returns a noisy sample of the similarity between two elements in $V$, to obtain a clustering of $V$ that minimizes the disagreements. Specifically, they introduced two novel online-learning problems rooted in the paradigm of combinatorial multi-armed bandits, and designed algorithms that combine KWIKCLUSTER with adaptive sampling strategies.

# B  OMITTED CONTENTS IN SECTION 4

## B.1  PROOF OF LEMMA 1

*Proof.* Fix $t \in \{1, \ldots, |\mathcal{B}|\}$. For simplicity, for any $r \in [0, 1/c]$, we write $B_{V^{(t)}}(i^{(t)}, r) = B(r)$, and moreover, for any $\ell \in [L]$, $\mathrm{cut}_{(V^{(t)}, \ell)}(B_{V^{(t)}}(i^{(t)}, r)) = \mathrm{cut}_\ell(r)$ and $\mathrm{vol}_{(V^{(t)}, \ell)}(B_{V^{(t)}}(i^{(t)}, r)) = \mathrm{vol}_\ell(r)$. By the definition of $r^*_{(t)}$, it suffices to show that there exists $r \in (0, 1/c]$ that satisfies

$$\max_{\ell \in [L]: F_\ell \neq 0} \frac{\mathrm{cut}_\ell(r)}{\mathrm{vol}_\ell(r)} \leq cL \log(n+1).$$

Suppose, for contradiction, that for any $r \in (0, 1/c]$,

$$\max_{\ell \in [L]: F_\ell \neq 0} \frac{\mathrm{cut}_\ell(r)}{\mathrm{vol}_\ell(r)} > cL \log(n+1).$$

Then we have

$$\int_0^{1/c} \max_{\ell \in [L]: F_\ell \neq 0} \frac{\mathrm{cut}_\ell(r)}{\mathrm{vol}_\ell(r)} \, \mathrm{d}r > \int_0^{1/c} cL \log(n+1) \, \mathrm{d}r = L \log(n+1). \tag{4}$$

Now relabel the elements in $V^{(t)}$ that have distance less than $1/c$ from $i^{(t)}$ (including $i^{(t)}$ itself) as $i^{(t)} = j_0, \ldots, j_{q-1}$ in the increasing order of the distance. For each $p = 0, \ldots, q-1$, we denote by $r_p$ the distance from $i^{(t)}$ to $j_p$, i.e., $r_p = x^*_{i^{(t)} j_p}$. For convenience, we set $r_q = 1/c$. For any $\ell \in [L]$, the function $\mathrm{vol}_\ell(r)$ is not necessarily differentiable and even not necessarily continuous at $r_0, \ldots, r_q$. On the other hand, at any point $r \in (0, 1/c]$ except for $r_1, \ldots, r_q$, the function $\mathrm{vol}_\ell(r)$ is differentiable, and from the definition, we have

$$\frac{\mathrm{d}\,\mathrm{vol}_\ell(r)}{\mathrm{d}r} = \mathrm{cut}_\ell(r). \tag{5}$$

Moreover, by simple calculation, we have that for any $\ell \in [L]$ with $F_\ell \neq 0$,

$$\frac{\mathrm{vol}_\ell(1/c)}{\mathrm{vol}_\ell(0)} \leq n+1. \tag{6}$$

Indeed, we see that $\mathrm{vol}_\ell(0) = F_\ell/n$ and

$$\mathrm{vol}_\ell(1/c) = \frac{F_\ell}{n} + \sum_{\{j,k\} \in E_\ell^+ : j, k \in B(1/c)} w_\ell(j,k) x^*_{jk} + \sum_{\{j,k\} \in E_\ell^+ : j \in B(1/c) \wedge k \in V^{(t)} \setminus B(1/c)} w_\ell(j,k) \left( \frac{1}{c} - x^*_{i^{(t)} j} \right)$$

$$\leq \frac{F_\ell}{n} + \sum_{\{j,k\} \in E_\ell^+ : j \in B(1/c) \wedge k \in V^{(t)}} w_\ell(j,k) x^*_{jk}$$

$$\leq \frac{F_\ell}{n} + F_\ell,$$

where the first inequality follows from

$$1/c - x^*_{i^{(t)} j} \leq x^*_{i^{(t)} k} - x^*_{i^{(t)} j} \leq x^*_{i^{(t)} j} + x^*_{jk} - x^*_{i^{(t)} j} = x^*_{jk} \tag{7}$$

for any $\{j, k\} \in E_\ell^+$ such that $j \in B(1/c)$ and $k \in V^{(t)} \setminus B(1/c)$. Using Equality (5) and Inequality (6), we have

$$
\begin{aligned}
\int_0^{1/c} \max_{\ell \in [L]:\, F_\ell \neq 0} \frac{\mathrm{cut}_\ell(r)}{\mathrm{vol}_\ell(r)} \, \mathrm{d}r &\leq \sum_{\ell \in [L]:\, F_\ell \neq 0} \int_0^{1/c} \frac{\mathrm{cut}_\ell(r)}{\mathrm{vol}_\ell(r)} \, \mathrm{d}r \\
&= \sum_{\ell \in [L]:\, F_\ell \neq 0} \sum_{p=0}^{q-1} \int_{r_p}^{r_{p+1}} \frac{\mathrm{cut}_\ell(r)}{\mathrm{vol}_\ell(r)} \, \mathrm{d}r \\
&= \sum_{\ell \in [L]:\, F_\ell \neq 0} \sum_{p=0}^{q-1} \int_{r_p}^{r_{p+1}} \frac{1}{\mathrm{vol}_\ell(r)} \, \mathrm{d}\, \mathrm{vol}_\ell(r) \\
&= \sum_{\ell \in [L]:\, F_\ell \neq 0} \sum_{p=0}^{q-1} (\log \mathrm{vol}_\ell(r_{p+1}) - \log \mathrm{vol}_\ell(r_p)) \\
&= \sum_{\ell \in [L]:\, F_\ell \neq 0} \log \frac{\mathrm{vol}_\ell(1/c)}{\mathrm{vol}_\ell(0)} \\
&\leq L \log(n+1),
\end{aligned}
$$

where the first inequality follows from the fact that $\frac{\mathrm{cut}_\ell(r)}{\mathrm{vol}_\ell(r)}$ is nonnegative for any $\ell \in [L]$ with $F_\ell \neq 0$ and $r \in (0, 1/c)$. The above contradicts Inequality (4), meaning that there exists $r \in (0, 1/c]$ such that

$$
\max_{\ell \in [L]:\, F_\ell \neq 0} \frac{\mathrm{cut}_\ell(r)}{\mathrm{vol}_\ell(r)} \leq cL \log(n+1).
$$

From now on, we show that $B(r_{(t)}^*) = B_{V^{(t)}}(i^{(t)}, r_{(t)}^*)$ can be computed in $O(Ln^2)$ time. To this end, it suffices to show that the radius $r_{(t)}^* \in \mathrm{argmin}\left\{\max_{\ell \in [L]:\, F_\ell \neq 0} \frac{\mathrm{cut}_\ell(r)}{\mathrm{vol}_\ell(r)} : r \in (0, 1/c]\right\}$ can be computed in $O(Ln^2)$ time. Recall the relabeling of the elements in $V^{(t)}$. For any $p = 0, \ldots, q-1$, in the interval $(r_p, r_{p+1}]$, the function $\frac{\mathrm{cut}_\ell(r)}{\mathrm{vol}_\ell(r)}$ for any $\ell \in [L]$ is monotonically nonincreasing, and thus so is $\max_{\ell \in [L]} \frac{\mathrm{cut}_\ell(r)}{\mathrm{vol}_\ell(r)}$. Indeed, in that interval, $\mathrm{cut}_\ell(r)$ is unchanged, while $\mathrm{vol}_\ell(r)$ is monotonically nondecreasing. Therefore, it suffices to compute $\max_{\ell \in [L]} \frac{\mathrm{cut}_\ell(r)}{\mathrm{vol}_\ell(r)}$ for all $r = r_1, \ldots, r_q$ and identify the one that attains the minimum. For each $\ell \in [L]$, we can compute $\mathrm{cut}_\ell(r)$ for all $r = r_1, \ldots, r_q$ in $O(n^2)$ time by iteratively moving the corresponding element and its incident edges. We can also compute $\mathrm{vol}_\ell(r)$ for all $r = r_1, \ldots, r_q$ in $O(n^2)$ time in a similar way. Performing these operations for all layers and computing the desired radius that attains the minimum requires $O(Ln^2)$ time. $\quad\square$

### B.2 PROOF OF THEOREM 1

*Proof.* By Lemma 1, each iteration of the region growing part (i.e., the while-loop) of Algorithm 1 can be performed in $O(Ln^2)$ time. As each iteration removes at least one element from the current set, the number of iterations is upper bounded by $n$. Therefore, we can obtain the time complexity presented in the theorem.

In what follows, we analyze the approximation ratio. Letting $\mathcal{B}$ be the output of the algorithm, we need to evaluate

$\|\text{Disagree}_\ell(\mathcal{B})\|_p$

$$
= \begin{cases} \left( \sum_{\ell \in [L]} \left( \sum_{\{j,k\} \in E_\ell^+} w_\ell(j,k) \mathbb{1}[\mathcal{B}(j) \neq \mathcal{B}(k)] + \sum_{\{j,k\} \in E_\ell^-} w_\ell(j,k) \mathbb{1}[\mathcal{B}(j) = \mathcal{B}(k)] \right)^p \right)^{1/p} \\ \qquad\qquad\qquad\qquad\qquad\qquad\qquad\qquad\qquad\qquad\qquad\qquad \text{if } p < \infty, \\[2ex] \max_{\ell \in [L]} \left( \sum_{\{j,k\} \in E_\ell^+} w_\ell(j,k) \mathbb{1}[\mathcal{B}(j) \neq \mathcal{B}(k)] + \sum_{\{j,k\} \in E_\ell^-} w_\ell(j,k) \mathbb{1}[\mathcal{B}(j) = \mathcal{B}(k)] \right) \\ \qquad\qquad\qquad\qquad\qquad\qquad\qquad\qquad\qquad\qquad\qquad\qquad \text{if } p = \infty. \end{cases}
$$

We first evaluate the terms for '+' labels. By Lemma 1, we have that for any $\ell \in [L]$ with $F_\ell \neq 0$,

$$
\text{cut}_{(V^{(t)}, \ell)}(B_{V^{(t)}}(i^{(t)}, r^*_{(t)})) \leq cL \log(n+1) \cdot \text{vol}_{(V^{(t)}, \ell)}(B_{V^{(t)}}(i^{(t)}, r^*_{(t)})).
$$

Based on this, for any $\ell \in [L]$ with $F_\ell \neq 0$, we have

$$
\sum_{\{j,k\} \in E_\ell^+} w_\ell(j,k) \mathbb{1}[\mathcal{B}(j) \neq \mathcal{B}(k)] = \sum_{t=1}^{|\mathcal{B}|} \text{cut}_{(V^{(t)}, \ell)}(B_{V^{(t)}}(i^{(t)}, r^*_{(t)}))
$$

$$
\leq cL \log(n+1) \sum_{t=1}^{|\mathcal{B}|} \text{vol}_{(V^{(t)}, \ell)}(B_{V^{(t)}}(i^{(t)}, r^*_{(t)}))
$$

$$
\leq cL \log(n+1) \left( \frac{F_\ell}{n} \cdot |\mathcal{B}| + \sum_{\{j,k\} \in E_\ell^+} w_\ell(j,k) x^*_{jk} \right)
$$

$$
\leq 2cL \log(n+1) \sum_{\{j,k\} \in E_\ell^+} w_\ell(j,k) x^*_{jk}. \tag{8}
$$

The second inequality follows from the fact that the balls included in $\mathcal{B}$ are mutually disjoint. Indeed, for any $\{j,k\} \in E_\ell^+$ contained in some ball $B_{V^{(t)}}(i^{(t)}, r^*_{(t)})$, the value $w_\ell(j,k) x^*_{jk}$ is produced just once due to $\text{vol}_{(V^{(t)}, \ell)}(B_{V^{(t)}}(i^{(t)}, r^*_{(t)}))$, while for any $\{j,k\} \in E_\ell^+$ across distinct balls $B_{V^{(t')}}(i^{(t')}, r^*_{(t')})$ and $B_{V^{(t'')}}(i^{(t'')}, r^*_{(t'')})$ $(t' < t'')$, once removing $B_{V^{(t')}}(i^{(t')}, r^*_{(t')})$, all the incident edges will never appear in the later iterations, and thus at most the value $w_\ell(j,k)(1/c - x^*_{i^{(t')}j})$ is produced just once due to $\text{vol}_{(V^{(t')}, \ell)}(B_{V^{(t')}}(i^{(t')}, r^*_{(t')}))$. Note that without loss of generality, we assumed that $B_{V^{(t')}}(i^{(t')}, r^*_{(t')})$ contains only $j$ among $j, k$. By Inequality (7), we have $1/c - x^*_{i^{(t')}j} \leq x^*_{jk}$. On the other hand, for any $\ell \in [L]$ with $F_\ell = 0$, we see that $x^*_{jk} = 0$ for any $\{j,k\} \in E_\ell^+$. Therefore, by its design, the algorithm does not separate any $\{j,k\} \in E_\ell^+$, meaning that for any $\ell \in [L]$ with $F_\ell = 0$,

$$
\sum_{\{u,v\} \in E_\ell^+} w_\ell(u,v) \mathbb{1}[\mathcal{B}(u) \neq \mathcal{B}(v)] = 0. \tag{9}
$$

Next we evaluate the terms for '−' labels. For any $\ell \in [L]$, we have

$$
\sum_{\{j,k\} \in E_\ell^-} w_\ell(j,k) \mathbb{1}[\mathcal{B}(j) = \mathcal{B}(k)] = \frac{c}{c-2} \sum_{t=1}^{|\mathcal{B}|} \sum_{\{j,k\} \in E_\ell^- : j,k \in B_{V^{(t)}}(i^{(t)}, r^*_{(t)})} w_\ell(j,k) \left( 1 - \frac{2}{c} \right)
$$

$$
\leq \frac{c}{c-2} \sum_{t=1}^{|\mathcal{B}|} \sum_{\{j,k\} \in E_\ell^- : j,k \in B_{V^{(t)}}(i^{(t)}, r^*_{(t)})} w_\ell(j,k) \left( 1 - x^*_{jk} \right)
$$

$$
\leq \frac{c}{c-2} \sum_{\{j,k\} \in E_\ell^-} w_\ell(j,k) \left( 1 - x^*_{jk} \right), \tag{10}
$$

where the first inequality follows from the triangle inequalities in (CV) and (LP). Indeed, denoting by $i^{(t)}$ the center of the ball containing $j, k$, we have $x_{jk}^* \leq x_{ji^{(t)}}^* + x_{i^{(t)}k}^* < 2/c$.

Let OPT be the optimal value of Problem 1. Using Inequality (8), Equality (9), and Inequality (10), we have that in the case of $p < \infty$,

$$\|\mathsf{Disagree}_\ell(\mathcal{B})\|_p$$

$$\leq \left( \sum_{\ell \in [L]} \left( 2cL \log(n+1) \sum_{\{j,k\} \in E_\ell^+} w_\ell(j,k) x_{jk}^* + \frac{c}{c-2} \sum_{\{j,k\} \in E_\ell^-} w_\ell(j,k) \left(1 - x_{jk}^*\right) \right)^p \right)^{1/p}$$

$$\leq \max\left\{ 2cL \log(n+1), \frac{c}{c-2} \right\} \left( \sum_{\ell \in [L]} \left( \sum_{\{j,k\} \in E_\ell^+} w_\ell(j,k) x_{jk}^* + \sum_{\{j,k\} \in E_\ell^-} w_\ell(j,k) \left(1 - x_{jk}^*\right) \right)^p \right)^{1/p}$$

$$= \max\left\{ 2cL \log(n+1), \frac{c}{c-2} \right\} \mathrm{OPT}_{\mathrm{CV}}$$

$$\leq \max\left\{ 2cL \log(n+1), \frac{c}{c-2} \right\} \mathrm{OPT},$$

and in the case of $p = \infty$,

$$\|\mathsf{Disagree}_\ell(\mathcal{B})\|_p$$

$$\leq \max_{\ell \in [L]} \left( 2cL \log(n+1) \sum_{\{j,k\} \in E_\ell^+} w_\ell(j,k) x_{jk}^* + \frac{c}{c-2} \sum_{\{j,k\} \in E_\ell^-} w_\ell(j,k) \left(1 - x_{jk}^*\right) \right)$$

$$\leq \max\left\{ 2cL \log(n+1), \frac{c}{c-2} \right\} \max_{\ell \in [L]} \left( \sum_{\{j,k\} \in E_\ell^+} w_\ell(j,k) x_{jk}^* + \sum_{\{j,k\} \in E_\ell^-} w_\ell(j,k) \left(1 - x_{jk}^*\right) \right)$$

$$= \max\left\{ 2cL \log(n+1), \frac{c}{c-2} \right\} \mathrm{OPT}_{\mathrm{LP}}$$

$$\leq \max\left\{ 2cL \log(n+1), \frac{c}{c-2} \right\} \mathrm{OPT}.$$

Noting that $\max\left\{ 2cL \log(n+1), \frac{c}{c-2} \right\} = O(L \log n)$, we have the theorem. $\qquad \square$

## C  OMITTED CONTENTS IN SECTION 5

### C.1  PROOF OF LEMMA 2

*Proof.* Fix $p \geq 1$. Let $V$ and $(w_\ell^+, w_\ell^-)_{\ell \in [L]}$ be the input of Problem 1 with the probability constraint, satisfying $w_\ell^+(u,v) + w_\ell^-(u,v) = 1$ for any $\ell \in [L]$ and $\{u,v\} \in E$. We construct an instance of Problem 2 as follows: Let $X = [0,1]^E$ and $d: X \times X \to \mathbb{R}_{\geq 0}$ be a metric such that $d(x,y) := \|x - y\|_1$ for $x, y \in X$. For $x \in X$ and $\{u,v\} \in E$, we denote by $x(u,v)$ the element of $x$ associated with $\{u,v\}$. For each $\ell \in [k]$, let $x_\ell \in X$ be the element such that $x_\ell(u,v) = w_\ell^-(u,v)$ for $\{u,v\} \in E$. Let $F = \{x \in \{0,1\}^E : x \text{ induces a clustering of } V\}$. Here $x$ is said to *induce a clustering of* $V$ if every connected component in $(V, E_x)$, where $E_x = \{\{u,v\} \in E : x(u,v) = 0\}$, is a clique. Then we see that there is a one-to-one correspondence between $F$ and the set of clusterings of $V$. Take an arbitrary element $x \in F$ and let $\mathcal{C}_x$ be the clustering corresponding to $x$. Then we have

that for any $\ell \in [L]$,

$$
\begin{aligned}
d(x, x_\ell) &= \|x - x_\ell\|_1 \\
&= \sum_{\{u,v\} \in E} \left( (1 - w_\ell^-(u,v)) \mathbb{1}[\mathcal{C}_x(u) \neq \mathcal{C}_x(v)] + w_\ell^-(u,v) \mathbb{1}[\mathcal{C}_x(u) = \mathcal{C}_x(v)] \right) \\
&= \sum_{\{u,v\} \in E} \left( w_\ell^+(u,v) \mathbb{1}[\mathcal{C}_x(u) \neq \mathcal{C}_x(v)] + w_\ell^-(u,v) \mathbb{1}[\mathcal{C}_x(u) = \mathcal{C}_x(v)] \right) \\
&= \mathsf{Disagree}_\ell(\mathcal{C}_x),
\end{aligned}
$$

meaning that the objective function of Problem 2 is equivalent to that of Problem 1 with the probability constraint. Therefore, $x$ is a $\beta$-approximate solution to Problem 2 if and only if so is $\mathcal{C}_x$ to Problem 1 with the probability constraint. Noticing that the above reduction can be done in polynomial time, we have the lemma. □

## C.2  PROOF OF THEOREM 2

*Proof.* Let $x^* \in F$ be an optimal solution to Problem 2. Let $x_{\mathrm{closest}} \in \operatorname{argmin}_{x \in \{x_1, \ldots, x_L\}} d(x, x^*)$ and $x'_{\mathrm{closest}}$ be the $\alpha$-approximate solution for Problem 3 with input $x_{\mathrm{closest}}$ and $F$. By the definition of $x'_{\mathrm{closest}}$ and $x_{\mathrm{closest}}$, we have that for any $\ell \in [L]$,

$$
d(x'_{\mathrm{closest}}, x_{\mathrm{closest}}) \leq \alpha \cdot d(x^*, x_{\mathrm{closest}}) \leq \alpha \cdot d(x^*, x_\ell).
$$

Using these inequalities, we have that for any $\ell \in [L]$,

$$
\begin{aligned}
d(x'_{\mathrm{closest}}, x_\ell) &\leq d(x'_{\mathrm{closest}}, x^*) + d(x^*, x_\ell) \\
&\leq d(x'_{\mathrm{closest}}, x_{\mathrm{closest}}) + d(x_{\mathrm{closest}}, x^*) + d(x^*, x_\ell) \\
&\leq \alpha \cdot d(x^*, x_\ell) + d(x^*, x_\ell) + d(x^*, x_\ell) \\
&= (\alpha + 2) \cdot d(x^*, x_\ell),
\end{aligned}
$$

where the first and second inequalities follow from the triangle inequality for the metric $d$ and the third inequality follows from the definition of $x_{\mathrm{closest}}$. Noticing that $x'_{\mathrm{closest}}$ is one of the output candidates of Algorithm 2, we can upper bound the objective value of the output $x_{\mathrm{out}}$ as follows: In the case of $p < \infty$,

$$
\left( \sum_{\ell \in [L]} d(x_{\mathrm{out}}, x_\ell)^p \right)^{1/p} \leq \left( \sum_{\ell \in [L]} d(x'_{\mathrm{closest}}, x_\ell)^p \right)^{1/p} \leq (\alpha + 2) \left( \sum_{\ell \in [L]} d(x^*, x_\ell)^p \right)^{1/p},
$$

while in the case of $p = \infty$,

$$
\max_{\ell \in [L]} d(x_{\mathrm{out}}, x_\ell) \leq \max_{\ell \in [L]} d(x'_{\mathrm{closest}}, x_\ell) \leq (\alpha + 2) \max_{\ell \in [L]} d(x^*, x_\ell),
$$

which concludes the proof. □

## C.3  PROOF OF COROLLARY 1

*Proof.* (i) By Lemma 2, it suffices to show that there exists a polynomial-time 4.5-approximation algorithm for Problem 2 with the metric space $(X, d)$ and the part of input $F \subseteq X$ that correspond to Problem 1 with the probability constraint. By Theorem 2, Algorithm 2 is an $(\alpha + 2)$-approximation algorithm for Problem 2, where $\alpha$ is the approximation ratio of the algorithm employed for solving Problem 3 with those $(X, d)$ and $F \subseteq X$. Based on the reduction in the proof of Lemma 2, Problem 3 with those $(X, d)$ and $F \subseteq X$ is equivalent to MINDISAGREE with the probability constraint, for which there exists a polynomial-time 2.5-approximation algorithm (Ailon et al., 2008). Therefore, we have the corollary.

(ii) The proof strategy is the same as the above. In this case, we can specialize the reduction given in the proof of Lemma 2 by replacing $X = [0, 1]^E$ with $X = \{0, 1\}^E$, and we see that Problem 3 with $(X, d)$ and $F \subseteq X$ is equivalent to MINDISAGREE of the unweighted case, for which there exists a polynomial-time $(1.437 + \epsilon)$-approximation algorithm for any $\epsilon > 0$ (Cao et al., 2024).

(iii) The proof is again similar. In this case, we can specialize the reduction by replacing $X = [0,1]^E$ with $X = \{x \in [0,1]^E : x(u,w) \leq x(u,v) + x(v,w), \ \forall u, v, w \in V\}$, and we see that Problem 3 with $(X,d)$ and $F \subseteq X$ is equivalent to MINDISAGREE with the probability constraint and the triangle inequality constraint, for which there exists a polynomial-time 1.5-approximation algorithm (Chawla et al., 2015). $\square$

### C.4 PROOF OF THEOREM 3

*Proof.* It suffices to prove that for any layer $\ell \in [L]$, it holds that

$$\mathsf{Disagree}_\ell(\mathcal{B}) \leq 4 \sum_{\{i,j\} \in E} \left( w_\ell^+(i,j) x_{ij}^* + w_\ell^-(i,j)(1 - x_{ij}^*) \right). \tag{11}$$

Indeed, from this inequality, it follows that $\|\mathbf{Disagree}(\mathcal{B})\|_p \leq 4 \cdot \mathrm{OPT}_{\mathrm{CV}}$ if $p < \infty$ and $\|\mathbf{Disagree}(\mathcal{B})\|_p \leq 4 \cdot \mathrm{OPT}_{\mathrm{LP}}$ if $p = \infty$, which proves the theorem. Fix $\ell \in [L]$ and consider an arbitrary iteration of the while-loop in Algorithm 3. Let $B \subseteq U$ be the cluster produced in the iteration. We define the *cost* of $B$ as the contribution of all pairs of elements in $U$ with at least one of them being inside $B$ to the objective value, i.e., $\sum_{\{j,k\} \in E: \, j,k \in B} w_\ell^-(j,k) + \sum_{\{j,k\} \in E: \, j \in B \wedge k \in U \setminus B} w_\ell^+(j,k)$. In what follows, we upper bound the cost of $B$ using the corresponding terms in the right-hand-side of Inequality (11). Recall that $C = B_U(i, 1/2) \setminus \{i\}$ contains all elements in $U$ (except for $i$) within distance of at most $1/2$ from $i$. There are two cases:

(i) If the average distance between $i$ and the elements in $C$ is no less than $1/4$, i.e., $\frac{1}{|C|} \sum_{j \in C} x_{ij}^* \geq 1/4$, then the algorithm forms the singleton cluster $B = \{i\}$. In this case, the cost of the cluster reduces to $\sum_{j \in U \setminus \{i\}} w_\ell^+(i,j)$. For each $j \in U \setminus \{i\}$ with $x_{ij}^* > 1/2$, we can upper bound each $w_\ell^+(i,j)$ using the corresponding term in the right-hand-side of Inequality (11) because it holds that $w_\ell^+(i,j) \leq 2 \cdot w_\ell^+(i,j) x_{ij}^* \leq 2 \left( w_\ell^+(i,j) x_{ij}^* + w_\ell^-(i,j)(1 - x_{ij}^*) \right)$. On the other hand, consider any pair of elements for which $x_{ij}^* \leq 1/2$ holds, i.e., the element $j$ is contained in $C$. Then, it holds that $1 - x_{ij}^* \geq x_{ij}^*$, and thus we have

$$\sum_{j \in C} \left( w_\ell^+(i,j) x_{ij}^* + w_\ell^-(i,j)(1 - x_{ij}^*) \right) \geq \sum_{j \in C} \left( w_\ell^+(i,j) + w_\ell^-(i,j) \right) x_{ij}^* = \sum_{j \in C} x_{ij}^*,$$

where the equality follows from the probability constraint. Using the above inequality together with the assumption $\frac{1}{|C|} \sum_{j \in C} x_{ij}^* \geq 1/4$, we have

$$\sum_{j \in C} w_\ell^+(i,j) \leq |C| \leq 4 \sum_{j \in C} x_{ij}^* \leq 4 \sum_{j \in C} \left( w_\ell^+(i,j) x_{ij}^* + w_\ell^-(i,j)(1 - x_{ij}^*) \right).$$

(ii) The second case is when the average satisfies $\frac{1}{|C|} \sum_{j \in C} x_{ij}^* < 1/4$, where the algorithm forms the cluster $B = \{i\} \cup C$. For the sake of the proof, we assume that the elements in $U$ are relabeled so that $j < k$ if $x_{ij}^* < x_{ik}^*$, where ties are broken arbitrarily.

First consider the pairs of elements contained in $B$. The cost of $B$ charged by these pairs is $\sum_{\{j,k\} \in E: \, j,k \in B} w_\ell^-(j,k)$. If both $x_{ij}^* < 3/8$ and $x_{ik}^* < 3/8$ hold, then the triangle inequality over the pseudometric assures that $1 - x_{jk}^* \geq 1/4$, and therefore each $w_\ell^-(j,k)$ can be upper bounded by the corresponding term in the right-hand-side of Inequality (11) within a factor of 4. The cost of $B$ charged by the remaining pairs of elements $j, k \in B$ with $j < k$ can be taken into account by $k$. Obviously we have $x_{ik}^* \in [3/8, 1/2]$. For a fixed $k$, define the quantities $p_k = \sum_{j<k} w_\ell^+(j,k)$ and $n_k = \sum_{j<k} w_\ell^-(j,k)$. The cost taken into account by $k$ is equal to $n_k$. The sum of the terms corresponding to all pairs $j < k$, where $k$ is fixed, in the right-hand-side of Inequality (11) can be lower bounded as follows:

$$\sum_{j<k} \left( w_\ell^+(j,k) x_{jk}^* + w_\ell^-(j,k)(1 - x_{jk}^*) \right) \geq \sum_{j<k} \left( w_\ell^+(j,k)(x_{ik}^* - x_{ij}^*) + w_\ell^-(j,k)(1 - x_{ik}^* - x_{ij}^*) \right)$$

$$= p_k x_{ik}^* + n_k(1 - x_{ik}^*) - \sum_{j<k} x_{ij}^*$$

$$\geq p_k x_{ik}^* + n_k(1 - x_{ik}^*) - \frac{p_k + n_k}{4}.$$

The last inequality follows from the probability constraint together with the fact that the average distance between $i$ and the elements in $\{j : j < k\}$ must be smaller than $1/4$, as $x_{ij}^* \geq 3/8$ for any $j \geq k$. Therefore, the above is lower bounded by a linear function depending on $x_{ik}^* \in [3/8, 1/2]$. It is easy to see that for every $x_{ik}^*$ in this range, the value is always at least $n_k/4$. Therefore, the cost $n_k$ is always within a factor of $4$.

Next consider the pairs of elements $j, k \in U$ with exactly one element being contained in $B = \{i\} \cup C$. Without loss of generality, we assume that $j < k$ and thus we have $j \in B$, $k \in U \setminus B$, and $x_{ij}^* < x_{ik}^*$. The cost of $B$ charged by these pairs is $\sum_{\{j,k\} \in E: \, j \in B \wedge k \in U \setminus B} w_\ell^+(j, k)$. If $x_{ik}^* \geq 3/4$ holds, then $x_{ik}^* - x_{ij}^* \geq 1/4$. Using the triangle inequality over the pseudometric, we have $x_{jk}^* \geq 1/4$, meaning that the cost charged by those pairs is accounted for within a factor of $4$. The cost of $B$ charged by the remaining pairs can again be taken into account by $k$. Obviously we have $x_{ik}^* \in (1/2, 3/4)$. For a fixed $k$, redefine the quantities $p_k = \sum_{j<k: \, j \in B} w_\ell^+(j, k)$ and $n_k = \sum_{j<k: \, j \in B} w_\ell^-(j, k)$. The cost taken into account by $k$ is equal to $p_k$. The rest of the proof is identical to the above. $\qquad\square$

# D OMITTED CONTENTS IN SECTION 6

Here we discuss Problem 1 with the probability constraint.

## D.1 EXPERIMENTAL SETUP

**Datasets.** The instances are generated with the same intuition as that for Problem 1 of the general weighted case. For each layer $\ell \in [L]$ and $\{u, v\} \in E$, if $\{u, v\} \in E_\ell$ we set $w_\ell^+(u, v) = 0.5 + w_\ell(\{u, v\})/2$ and $w_\ell^-(u, v) = 1 - w_\ell^+(u, v)$; otherwise we set $w_\ell^+(u, v) = 1 - w_\ell^-(u, v)$, where $w_\ell^-(u, v) = 0.5 + \texttt{random.choice(weights}(\ell))/2$ with probability $0.5$, and $w_\ell^+(u, v) = w_\ell^-(u, v) = 0.5$ otherwise.

**Our algorithms and baselines.** We run Algorithms 2 and 3. Note that Algorithm 2 varies depending on the approximation algorithm for MINDISAGREE with the probability constraint employed in the algorithm. Specifically, we use the 2.5-approximation algorithm and the 5-approximation algorithm, designed by Ailon et al. (2008), providing the approximation ratios of 4.5 and 7, respectively, of Algorithm 2. There is a trade-off between these two selections: The first algorithm has a better approximation ratio, but it is slower, as it has to solve an LP, which is not required in the second algorithm. We refer to the two algorithms as Algorithm 2 (LP) and Algorithm 2 ($\overline{\text{LP}}$), respectively. In Algorithm 3, the way to select a pivot is arbitrary, and we use the same rule as that for Algorithm 1. We employ the following baseline method, which we refer to as Aggregate-Pr. This method is the probability-constraint counterpart of Aggregate. Specifically, the method constructs $w^+ : E \to \mathbb{R}_{\geq 0}$ and $w^- : E \to \mathbb{R}_{\geq 0}$ by setting $w^+(u, v) = \left( \sum_{\ell \in [L]} w_\ell^+(u, v) \right) / L$ and $w^-(u, v) = \left( \sum_{\ell \in [L]} w_\ell^-(u, v) \right) / L$ for every $\{u, v\} \in E$. Then it solves MINDISAGREE with the probability constraint with input $V$ and $(w^+, w^-)$, using the 2.5-approximation algorithm or the 5-approximation algorithm (Ailon et al., 2008), as in Algorithm 2. We refer to this baseline as Aggregate-Pr (LP) or Aggregate-Pr ($\overline{\text{LP}}$), depending on the choice of the above approximation algorithm. As mentioned in Section 3, Aggregate-Pr (LP) gives a 2.5-approximate solution for Problem 1 with the probability constraint when $p = 1$, but the approximation ratio for the case of $p = \infty$ is not clear.

## D.2 RESULTS

The results are summarized in Tables 2 and 3 (just separated due to space constraints). Note that for this case, all algorithms except for Algorithm 3 are performed 10 times, as they contain randomness. OT again indicates that (the first run of) the algorithm did not terminate in 3,600 seconds. The objective values are presented using the average value and the standard deviation, while the running time is just with the average value, because obviously it may not vary much. The trend of the results is similar to that for the general weighted case. Indeed, Algorithm 3 with an approximation ratio of $4$ outperforms the baseline methods in terms of the quality of solutions, and the algorithm succeeds in obtaining near-optimal solutions. Although Algorithm 2 (LP) and Algorithm 2 ($\overline{\text{LP}}$) are also our

Table 2: Results for Problem 1 with the probability constraint.

| Dataset | LB | Algorithm 2 (LP) | | Algorithm 2 ($\overline{\text{LP}}$) | | Algorithm 3 | |
|---|---|---|---|---|---|---|---|
| | | Obj. val. | Time(s) | Obj. val. | Time(s) | Obj. val. | Time(s) |
| aves-sparrow-social | 630.8 | 635.1±1.8 | 0.4 | 658.0±1.6 | **0.0** | **631.1** | 0.3 |
| insecta-ant-colony1 | 3148.2 | 3154.5±0.5 | 1728.4 | 3160.7±1.2 | 1.0 | **3150.3** | 674.2 |
| reptilia-tortoise-network-bsv | 2387.5 | 2683.3±40.6 | 19.8 | 3837.7±54.6 | **0.0** | **2422.5** | 2.9 |
| aves-wildbird-network | 9840.2 | 9887.9±2.6 | 142.0 | 10077.8±8.4 | 0.1 | **9841.3** | 11.2 |
| aves-weaver-social | 24875.7 | — | OT | 39732.3±342.5 | 5.3 | **24924.5** | 94.1 |
| reptilia-tortoise-network-fi | 77569.5 | — | OT | 126849.1±831.5 | 3.2 | **77577.5** | 189.5 |

Table 3: Results for Problem 1 with the probability constraint (continued).

| Dataset | LB | Aggregate-Pr (LP) | | Aggregate-Pr ($\overline{\text{LP}}$) | |
|---|---|---|---|---|---|
| | | Obj. val. | Time(s) | Obj. val. | Time(s) |
| aves-sparrow-social | 630.8 | 638.1±1.7 | 0.1 | 652.7±2.1 | **0.0** |
| insecta-ant-colony1 | 3148.2 | 3154.0±0.1 | 60.3 | 3158.3±3.4 | **0.0** |
| reptilia-tortoise-network-bsv | 2387.5 | 2444.5±13.4 | 0.9 | 2601.0±18.2 | **0.0** |
| aves-wildbird-network | 9840.2 | 9863.2±4.7 | 6.2 | 9900.9±17.4 | **0.0** |
| aves-weaver-social | 24875.7 | 24971.5±0.0 | 10.3 | 24971.0±0.0 | **0.2** |
| reptilia-tortoise-network-fi | 77569.5 | 77664.7±5.1 | 123.5 | 77740.8±12.4 | **0.2** |

proposed algorithms, which have approximation ratios of 4.5 and 7, respectively, their practical performances are not comparable with that of Algorithm 3. Therefore, we conclude that our proposed algorithm for practical use is Algorithm 3.

# E OMITTED CONTENTS IN SECTION 7

## E.1 DETAILED DESCRIPTIONS OF OPEN PROBLEMS

For Problem 1 of the general weighted case, can we design a polynomial-time algorithm that has an approximation ratio better than $O(L \log n)$? As Problem 1 contains MINDISAGREE as a special case and approximating MINDISAGREE is known to be harder than approximating Minimum Multicut (Garg et al., 1996), it is quite challenging to obtain an approximation ratio of $o(\log n)$. Therefore, a more reasonable question is "to what extent can we reduce the term $L$ in the current approximation ratio of $O(L \log n)$?" To answer this, the first step would be to investigate the integrality gaps of (CV) and (LP). The current integrality gap of $\Omega(\log n)$, inherited from the LP relaxation used in the $O(\log n)$-approximation algorithms for MINDISAGREE (Charikar et al., 2005; Demaine et al., 2006), leaves the possibility to improve the approximation ratio of Algorithm 1 to $O(\log n)$. Another interesting direction is to improve the approximation ratios for Problem 1 with the probability constraint and its special cases. For instance, can we design a polynomial-time algorithm that has an approximation ratio better than 4 for the general case? To this end, one possibility is to improve the approximation ratio for MINDISAGREE with the probability constraint from the current best 2.5 (Ailon et al., 2008) to some value smaller than 2. As the integrality gap of the LP relaxation used in the 2.5-approximation algorithm (i.e., KWIKCLUSTER) is known to be 2 (Charikar et al., 2005), this approach requires to invent a different technique. Another possibility is to replace the rounding procedure of Algorithm 3 to that of KWIKCLUSTER, but it is not clear how to extend the analysis focusing on the *bad triplets* (Ailon et al., 2008) to the multilayer setting. For Problem 1 of the unweighted case and Problem 1 with the probability constraint and the triangle inequality constraint, improving the approximation ratio for the single-layer counterpart directly improves our approximation ratios. Finally, investigating Multilayer Correlation Clustering in the spirit of MAXAGREE rather than MINDISAGREE is also an interesting direction. It is worth mentioning that a closely-related problem called Simultaneous Max-Cut has recently been studied by Bhangale et al. (2018) and Bhangale & Khot (2020) from the approximability and inapproximability points of view, respectively.

