# OpenReview forum: "Multilayer Correlation Clustering"
_ICLR.cc/2025/Conference — Submitted to ICLR 2025_

### Official Review · Reviewer_LSph · 2024-11-01

**Soundness:** 3
**Presentation:** 3
**Contribution:** 3
**Rating:** 6
**Confidence:** 3

**Summary:**

The paper studies multilayer correlation clustering, which is a generalization of correlation clustering. Each layer represents a distinct correlation clustering instance on the same set of vertices. The goal is to find a consistent clustering for all layers while minimizing the p-norm of the multilayer-disagreements vector.

The main result is an $O(L\log n)$-approximation algorithm for multilayer correlation clustering and improved algorithms for the special case of the problem with a probability constraint. The authors provide theoretical proofs and experimental evaluations to demonstrate the effectiveness of the algorithms.

**Strengths:**

- The definition of "multilayer correlation clustering" looks natural, and the motivation is clear.

- Under the new model, the proposed algorithm achieves a good approximation ratio. In the case of probability constraints, the algorithm can achieve a constant approximation ratio.

- The paper is easy to read. The explanation of the convex programming problem and the algorithm is clear.

- The experimental results are good, obtaining near-optimal solutions in various real-world datasets.

**Weaknesses:**

- Lack of theoretical justification for the effectiveness of the results: There is no comparative analysis of the algorithm with related work (e.g., MCCC, Bonchi et al. (2015)), nor is there a lower bound provided.

- The approach requires to solve CV or LP which are heavy for larger datasets. (In the experiments, only $p = \infty$ was tested; I suspect that if other values of $p$ are used, the running time will be longer due to the need to solve CV.)

**Questions:**

- Only arXiv version is cited for several papers — are they published in a conference? Please check.

- The baselines (Pick-a-Best and Aggregate) are relatively trivial algorithms, but in the experiments, their results seem to also approach the optimal solution of the LP. Does this suggest that there may be some issues with the experimental setup?

---

> ### Author Response · Authors · 2024-11-22
>
> We greatly appreciate your careful reading and invaluable feedback.
>
> > Lack of theoretical justification for the effectiveness of the results: There is no comparative analysis of the algorithm with related work (e.g., MCCC, Bonchi et al. (2015)), nor is there a lower bound provided.
>
> Although the input of MCCC is essentially the same as that of our problem of the unweighted case, the objective function is completely different, as explained on Lines 155–161. Therefore, it is very unlikely that the existing algorithms for MCCC provide an approximation guarantee for our problem, even if we only consider the unweighted case. If we consider the general weighted case, it seems impossible, because MCCC does not take into account the weights and the way to generalize it to the weighted case is not trivial. Please note that as mentioned on Lines 43--45, improving the $O(\log n)$-approximation for MinDisagree, a quite special case of our problem, is at least as hard as improving the $O(\log n)$-approximation for Minimum Multicut (Garg et al., 1996), which is one of the major open problems in theoretical computer science. This is not a lower bound, but provides a certificate of difficulty of approximation.
>
> > The approach requires to solve CV or LP which are heavy for larger datasets. (In the experiments, only $p=\infty$ was tested; I suspect that if other values of $p$ are used, the running time will be longer due to the need to solve CV.)
>
> Yes, if other values of $p$ are used, the running time could be longer. However, the reason why we decided not to perform our experiments with other values of $p$ is based on the following facts:
>
> - CV is still solvable to arbitrary precision in polynomial time, but implementing the algorithms for the problem (e.g., the interior-point method (Boyd & Vandenberghe, 2004)) requires much effort, which is beyond the interest of our work focusing particularly on the theory.
>
> - For reference, let us mention the experimental settings of the papers that study the local objectives for MinDisagree, as the contexts in which this model and our model are situated are very similar. As reviewed in Appendix A.1, in the model with the local objectives, the disagreements of a clustering are quantified locally rather than globally, at the level of single elements. Specifically, the papers consider a disagreements vector (with dimension equal to the number of elements), where $i$-th element represents the disagreements incident to the corresponding element $i\in V$. The goal is then to minimize the $\ell_p$-norm ($p\geq 1$) of the disagreements vector. This model is totally different from our model, but in the sense that both of them deal with $\ell_p$-norms in the objective, they are similar. To the best of our knowledge, there exists no paper that performs experiments for the case of $p\in (1,\infty)$, though Davies et al. (2023) and Heidrich et al. (2024) conducted experiments for the case of $p=\infty$. Therefore, our experimental setting is consistent with those in the literature.
>
>
> > Only arXiv version is cited for several papers — are they published in a conference? Please check.
>
> In our reference list, we have two arXiv papers: Ahmadi et al. (2020) and Kuroki et al. (2024). We could not find any conference or journal version of Ahmadi et al. (2020), but we found Kuroki et al. (2024) accepted to NeurIPS 2024. In our revised version, we will update the information properly. Thank you for your question.
>
> > The baselines (Pick-a-Best and Aggregate) are relatively trivial algorithms, but in the experiments, their results seem to also approach the optimal solution of the LP. Does this suggest that there may be some issues with the experimental setup?
>
> Pick-a-Best and Aggregate do not approach the lower bound provided by the LP. For convenience, we summarize the results of Table 1 below. Indeed, the objective value achieved by Pick-a-Best is almost equal to twice the lower bound for the first and fourth dataset. Pick-a-Best works slightly better, but it usually achieves the objective value greater than 1.3 times the lower bound; in particular, for the last dataset, it achieves 1.64 times the lower bound. On the other hand, the objective value achieved by Algorithm 1 is often quite close to the lower bound, and even for the worst case (i.e., aves-weaver-social), the objective value is less than 1.24 times the lower bound.
>
> |Dataset|LB|Algorithm 1|Pick-a-Best|Aggregate|
> |:---|---:|---:|---:|---:|
> |aves-sparrow-social  | 13.37  | 13.48 |  26.79 | 13.81 |
> | insecta-ant-colony1  |32.48| 34.30 |    42.94    |         47.59 |
> |reptilia-tortoise-network-bsv | 127.14  |   151.00    |       193.00    |       174.00 |
> |aves-wildbird-network    |         54.97  |     56.50    |         98.27     |        74.84 |
> |aves-weaver-social       |        132.75  |    164.00    |             —     |       177.00 |
> |reptilia-tortoise-network-fi   |   271.48  |   305.00      |          —     |         446.00 |

---

> > ### Comment · Reviewer_LSph · 2024-11-26
> >
> > Thanks for your responses. My initial assessment still holds.

---

### Official Review · Reviewer_LcEF · 2024-11-01

**Soundness:** 2
**Presentation:** 3
**Contribution:** 2
**Rating:** 3
**Confidence:** 3

**Summary:**

The authors propose a generalization of correlation clustering problem, termed as multilayer correlation clustering. In addition, polynomial time $O(L \log(n) )$-accurate approximation algorithms are proposed to solve the generalized problem. The main idea is to relax the original problem to a convex problem.

**Strengths:**

The main contribution is to propose a polynomial time algorithm to output a solution of $O(L \log(n))$ accuracy for the generalized correlation clustering problem.

**Weaknesses:**

1, The proposed multilayer correlation clustering problem lacks motivations. The aggregation of information from multiple weight functions $w_{l}^{+}$, $w_{l}^{-}, l=1,2, \dots, L$ can be done through more convenient and efficient ways. For instance, one can aggregate information by aggregating the weight functions by considering $w^{+} = \max_l w_{l}^{+}$,  $w^{-} = \max_l w_{l}^{-}$ or  $w^{+} = \sum_l w_{l}^{+}$, $w^{-} = \sum_l w_{l}^{-}$ or $w^{+} = (\sum_l (w_{l}^{+})^p)^{1/p}$, $w^{-} = (\sum_l (w_{l}^{-})^p)^{1/p}$. The benefits of solving the multilayer correlation clustering problem are not be well-established in the paper.

2, In section 6.1, since the case $p=\infty$ is considered, it's fairer to compare with aggregated functions $w^{+} = \max_l w_{l}^{+}$,  $w^{-} = \max_l w_{l}^{-}$.

3, Two baseline optimization methods for solving problem 1 are compared in the simulations. However, a method for comparing information gain and clustering accuracy for problem 1 is currently lacking.

**Questions:**

What additional information can be gained from multilayer correlation clustering compared to simply utilizing aggregated weight functions?

---

> ### Author Response · Authors · 2024-11-22
>
> We greatly appreciate your careful reading and invaluable feedback.
>
> > W1
>
> Regarding the motivations, please read Lines 63–73 in our manuscript: Multilayer Correlation Clustering is motivated by real-world scenarios. Suppose that we want to find a clustering of users of $\mathbb{X}$ using their similarity information. In this case, various types of similarity can be defined through analysis of users' tweets and observations of different types of connections among users such as follower relations, retweets, and mentions. In the original Correlation Clustering, we need to deal with that information one by one and manage to aggregate resulting clusterings. On the other hand, Multilayer Correlation Clustering enables us to handle that information simultaneously, directly producing a clustering that is consistent (as much as possible) with all types of information. As another example scenario, suppose that we analyze brain networks, where nodes correspond to small regions of a brain and edges represent similarity relations among them. Then it is often the case that the edge set is not determined uniquely; indeed, there would be at least two types of similarity based on the structural connectivity and the functional connectivity among the small pieces of a brain. Obviously, Multilayer Correlation Clustering can again find its advantage in this context.
>
> Your suggested aggregation rules of weight functions over layers are not well-defined. For instance, the reviewer suggests $w^+:= \max\_{\ell \in [L]} w^+\_\ell$ and $w^-:= \max\_{\ell \in [L]} w^-\_\ell$; however, as $w^+\_\ell$ and $w^-\_\ell$ are functions, the max-operator over $\ell\in [L]$ is not well-defined. Instead, we could consider aggregation rules such as $w^+(u,v):= \max\_{\ell \in [L]} w^+\_\ell(u,v)$ and $w^-(u,v):= \max_{\ell \in [L]} w^-\_\ell(u,v)$ for $\\{u,v\\}\in E$.
>
> Of course, aggregating different layers into one layer, and then solving MinDisagree on only one layer, is an efficient attempt at solving our problem, and this is exactly what the baseline Aggregate does. However, as demonstrated by the example below, and in our experiments, such methods perform very poorly with respect to approximating the objective function. Please note that one of the suggested aggregation rules is exactly the same as that employed in the baseline Aggregate.
>
> As in the definition, the objective of Multilayer Correlation Clustering is to find a clustering that minimizes the $\ell_p$-norm of the multilayer-disagreement vector. In particular, as the most intuitive case, if we set $p=\infty$, we can minimize the maximum disagreement over layers. This is the clear benefit of solving Multilayer Correlation Clustering.
>
> > W2
>
> Please notice that the suggested aggregation rule does not perform well. Intuitively speaking, for each pair of elements, the rule forgets about anything except for the maximum weight, resulting in a poor performance even in the case of $p=\infty$. We can prove this theoretically. Consider the following instance: Fix $p=\infty$. The set $V$ consists of $n$ elements. There are ${n\choose 2}$ layers, where in each layer there is only one pair having \`$+$’ label with weight $1$ and all the other ${n\choose 2} - 1$ pairs have \`$-$’ label with weight $1$, such that the pairs having `$+$’ label with weight $1$ are distinct over the layers. Then the optimal solution to this instance is a clustering consisting of $n$ singleton clusters, achieving the optimal value being just $1$ for Problem 1. On the other hand, the suggested aggregation rule produces $w^+(u,v)= 1$ and $w^-(u,v)=1$ for all $\{u,v\}\in E$, and it is obvious that the optimal solution to MinDisagree with these weights can be any clustering, e.g., the clustering consisting of a single cluster containing all elements, achieving the objective value being ${n\choose 2}-1$ for Problem 1. Therefore, the approximation ratio of this algorithm is $\Omega(n^2)$. On the other hand, the above instance is an instance of Problem 1 of the unweighted case, and therefore, our proposed algorithm, i.e., Algorithm 2, achieves the approximation ratio of $3.437+\epsilon$, for any $\epsilon >0$.
>
> > W3
>
> It is not clear what the information gain for Problem 1 means. If the reviewer could provide its exact definition, we would be open to discuss it. On the other hand, as Problem 1 is an optimization problem, the clustering accuracy should be measured using the objective function, which is exactly what we did in the experiments. The objective value (the lower the better) achieved by Algorithm 1 is often quite close to the lower bound (and thus the optimal value).
>
> > What additional information...?
>
> As mentioned above, Multilayer Correlation Clustering provides additional information, e.g., a clustering that minimizes the maximum agreements over layers. As proved above, simply aggregating weight functions works very poorly to solve the problem, highlighting the effectiveness of our proposed algorithms.

---

> > ### Comment · Reviewer_LcEF · 2024-11-26
> > **Response to Authors' Comments**
> >
> > Thanks for the comments.
> >
> > You wrote "Of course, aggregating different layers into one layer, and then solving MinDisagree on only one layer, is an efficient attempt at solving our problem, and this is exactly what the baseline Aggregate does. However, as demonstrated by the example below, and in our experiments, such methods perform very poorly with respect to approximating the objective function."
> >
> > Is the objective function of the clustering results of baseline Aggregate computed by setting $p=\infty$? If so, this is not appropriate for comparing your approach (p>1) with the baseline case (p=1). The baseline (p=1) is just your approach by setting (p=1).

---

> > > ### Author Response · Authors · 2024-11-28
> > >
> > > Thank you very much for your active participation during the discussion period.
> > >
> > > > Is the objective function of the clustering results of baseline Aggregate computed by setting $p=\infty$? If so, this is not appropriate for comparing your approach (p>1) with the baseline case (p=1). The baseline (p=1) is just your approach by setting (p=1).
> > >
> > > Yes, as stated in the beginning of Section 6.1, we set $p=\infty$ in Problem 1 throughout the experiments; thus, the objective function is with $p=\infty$. However, we do not think that the aggregation rule of $w^+(u,v):= \max\_{\ell \in [L]} w^+\_\ell(u,v)$ and $w^-(u,v):= \max\_{\ell \in [L]} w^-\_\ell(u,v)$, which we call the aggregation rule with $p=\infty$, is more appropriate than that of $w^+(u,v):= \sum\_{\ell \in [L]} w^+\_\ell(u,v)$ and $w^-(u,v):= \sum\_{\ell \in [L]} w^-\_\ell(u,v)$, which we call the aggregation rule with $p=1$, as a baseline. Indeed, as in the aggregation rule with $p=1$, the aggregation rule with $p=\infty$ also breaks the layer-wise information, and therefore, it is not clear if the aggregation rule with $p=\infty$ is more effective for minimizing the objective function even for the case of $p=\infty$. Please also note that for the instance we used in our rebuttal to prove that the aggregation rule with $p = \infty$ only has an approximation ratio of \Omega(n^2), the aggregation rule with $p = 1$ outputs an optimal solution.
> > >
> > > However, to honor the reviewer’s suggestion, we have implemented and tested the aggregation rule with $p=\infty$, which we call Aggregate (new). The objective values of solutions are presented in the following table. As shown, Aggregate (new) does not generally perform better than Aggregate. Although for aves-wildbird-network, Aggregate (new) performs slightly better, for insecta-ant-colony1 and reptilia-tortoise-network-fi, Aggregate (new) performs worse. In particular, the objective value for insecta-ant-colony1 is more than 25% worse than that achieved by Aggregate. In our revised version, we can report the results.
> > >
> > > | Dataset | LB | Algorithm 1 | Pick-a-Best | Aggregate | Aggregate (new) |
> > > |:---|---:|---:|---:|---:|---:|
> > > | aves-sparrow-social | 13.37 | 13.48 | 26.79 | 13.81 | 13.81 |
> > > | insecta-ant-colony1 | 32.48 | 34.30 | 42.94 | 47.59 | 59.97 |
> > > | reptilia-tortoise-network-bsv | 127.14 | 151.00 | 193.00 | 174.00 | 174.00 |
> > > | aves-wildbird-network | 54.97 | 56.50 | 98.27 | 74.84 | 68.76 |
> > > | aves-weaver-social | 132.75 | 164.00 |  —  | 177.00 | 177.00 |
> > > | reptilia-tortoise-network-fi | 271.48 | 305.00 | — | 446.00 | 482.00 |

---

> > > > ### Comment · Reviewer_LcEF · 2024-11-28
> > > > **Response to Authors' Comments**
> > > >
> > > > Thank you. How do you set $p$ in algorithm 1?

---

> > > > > ### Author Response · Authors · 2024-11-28
> > > > >
> > > > > Thank you for your response. Please note that there is no room for setting $p$ in Algorithm 1. It is determined by the problem itself. Again, in the experiments, we set $p=\infty$ in Problem 1, and thus (LP) is solved in Algorithm 1.

---

> ### Comment · Reviewer_LcEF · 2024-11-28
> **Response to Authors' Comments**
>
> Thank you. As a summary:
>
> 1, Problem 1 with $p=1$ is equivalent to MinDisagree with dissimilarity measure $w^{+}(u,v) = \sum_{l=1}^L w^{+}(u,v)$ (proposed in literature), which is the baseline approach `Aggregate` in the numerical studies.
>
> 2, Problem 1 with $p=\infty$ is the method `algorithm 1` in numerical studies, which may NOT be equivalent to MinDisagree with $w^{+}(u,v) = \max_{l} w^{+}(u,v)$. MinDisagree with $w^{+}(u,v) = \max_{l} w^{+}(u,v)$ is the new added method `Aggregate (new)` in numerical studies.
>
> The clustering results of `algorithm 1` and `Aggregate` are evaluated by computing Problem 1 with $p=\infty$.  The clustering results of `algorithm 1` are obtained by minimizing Problem 1 with $p=\infty$, of course, it will win. This comparison is insufficient, I keep my ratings.

---

> > ### Author Response · Authors · 2024-11-30
> >
> > Thank you for your response. However, we respectfully point out that the reviewer’s conclusion is inconsistent with their earlier comments.
> >
> > Specifically, the reviewer initially emphasized that it would be "fairer and more appropriate" to compare our proposed algorithm with a new baseline, where the aggregation rule is modified from $w^+(u,v):= \sum\_{\ell \in [L]} w^+\_\ell(u,v)$ and $w^-(u,v):= \sum\_{\ell \in [L]} w^-\_\ell(u,v)$ to $w^+(u,v):= \max\_{\ell \in [L]} w^+\_\ell(u,v)$ and $w^-(u,v):= \max\_{\ell \in [L]} w^-\_\ell(u,v)$. Although we theoretically justified in our rebuttal why we do not consider this modification to be a fairer or more appropriate baseline, we still implemented and tested this suggested baseline, which we referred to as Aggregate (new). Our results demonstrated that Aggregate (new) does not generally perform better than the original baseline Aggregate and, moreover, performs significantly worse than our proposed algorithm.
> >
> > Despite these findings, the reviewer’s final assessment disregards Aggregate (new) entirely and bases their conclusion solely on a comparison between our proposed algorithm and the original baseline Aggregate. This approach overlooks the explicit inclusion of Aggregate (new), which was introduced at the reviewer’s request and provides critical context for understanding the relative practical performance of the algorithms.
> >
> > We are sure that considering all baselines, including Aggregate (new), is essential for ensuring the consistency of the reviewer’s assessment. By excluding this baseline, the conclusion drawn does fail to reflect the evidence provided. We respectfully request that the evaluation be revisited with these points in mind.

---

> ### Comment · Reviewer_LcEF · 2024-11-30
> **Response to Authors' Comments**
>
> You wrote ``Despite these findings, the reviewer’s final assessment disregards Aggregate (new) entirely and bases their conclusion solely on a comparison between our proposed algorithm and the original baseline Aggregate."
>
> In my opinion, the comparison between algorithm 1 (problem 1 with $p=\infty$) and Aggregate (new) is also inadequate for the same reason: the clustering results are evaluated by computing problem 1 with $p=\infty$, algorithm 1 wins for sure.

---

> > ### Author Response · Authors · 2024-12-01
> >
> > Thank you for your response. We find it quite surprising that the reviewer has suddenly introduced a new perspective, asserting that comparing our proposed algorithm with Aggregate (new) is also inadequate. This perspective contradicts the reviewer’s earlier suggestion that Aggregate (new) represents a "fairer and more appropriate" baseline for comparison.
> >
> > If the reviewer’s position is that our experiments are inadequate due to the selection of baselines, we respectfully believe that it is incumbent upon the reviewer to suggest a baseline that they consider adequate for a comparison. However, we would like to emphasize that the objective value achieved by our proposed algorithm is often very close to the lower bound and hence the optimal value on the tested instances, ensuring that outperforming our proposed algorithm is fundamentally infeasible.

---

### Official Review · Reviewer_USqv · 2024-11-04

**Soundness:** 3
**Presentation:** 3
**Contribution:** 3
**Rating:** 5
**Confidence:** 4

**Summary:**

In classical correlation clustering, we have a complete graph, where every edge is either labeled + or -. And there are non-negative weights on edges. The goal is to cluster the nodes into parts, so that the total disagreement is minimized. That is, total weight of + edges going between parts, and total weight of - edges going inside parts.
General problem  admit O(log n) approximation using region growing approach, and special case when weights are all 1, admits O(1) approximation ratio, which has been improved over many prior work.

This paper studies a new generalization. Imagine we have L such graphs with weights and labels, and wish to find a "single" clustering which works well for all L graphs. How to aggregate the scores? let (D_1, D_2, .., D_L) denote the L scores for a given clustering. Then we can consdier the l_p norm of this vector to be the quantitiy we are trying to optimize.

Paper also studies one more "probability" version, where each edge has two weights w+ and w-, which add up to 1.

For the general problem, they show O(L log n) approximation factor.
For the probability version (where the weights add up to 1 for each layer), they show two results: one alpha+2 approximation and one 4-approximation, where alpha is the single-layer approximation factor.

**Strengths:**

Well written paper
Results are interesting from a theoretical perspective
Paper could spark nice follow-up work as it leaves many interesting challenges open
It is rare to find a theory paper run experiments of the kind this paper does, so much credit to the authors :)

**Weaknesses:**

Not sure how suitable the paper is for ICLR audience, as it is more of an SODA/ALENEX type paper in my humble opinion.
(Not taking anything away from the technical merits!)

In Section 5.1, Authors could do much more justice in explaining how they use Problem 2 to solve the general problem. In particular, what metric they use, what are x1, .., x_L and what is F? Are these the different solutions we get from the convex program? and metric space is the space of all solutions? Adding this details would make it more readable and interesting.  Also stress that Problem 3 is challenging only because the metric space could be huge.

6.1 Are there no real-world datasets without any semi-synthetic aspect to sampling the weights? Especially sampling negative weights from the positive weights.

Why is pick the best not run for the larger datasets?

Are there any other baselines one can think of? Perhaps some combination of adding the weights and pick a best? Perhaps using some approximation for l_p norm and then inferring a sampling strategy based on that? Like Multiplicative Weights method to weight the different layer instances?

**Questions:**

Line 135: "2.5 approximation, respectively" -- means what? this is unclear.
Line 138: Just to understand better, if the - weights satisfy triangle inequality, are the - label weights themselves positive values, or are the negative values?  This becomes clearer later, but was unclear at this point.

Line 231: "Note however that for Problem 1 of the unweighted case" -- what does this mean?

---

> ### Author Response · Authors · 2024-11-22
>
> We greatly appreciate your careful reading and invaluable feedback.
>
> > Not sure how suitable the paper is for ICLR audience, as it is more of an SODA/ALENEX type paper...
>
> Thank you for recognizing the technical merits of our work. According to the Call for Papers of ICLR 2025, the conference welcomes submissions from all areas of machine learning. Considering the fact that Correlation Clustering has actively been studied in the literature of machine learning, we believe that our work completely falls into the scope of the conference.
>
> > In Section 5.1, Authors could do much more justice in explaining how they use Problem 2...
>
> In our current manuscript, the descriptions are provided in the proof of Lemma 2, which can be found in Appendix C.1. Fix $p\geq 1$. Let $V$ and $(w^+\_\ell,w^-\_\ell)\_{\ell \in [L]}$ be the input of Problem 1 with the probability constraint, satisfying $w^+\_\ell(u,v)+w^-\_\ell(u,v)=1$ for any $\ell\in [L]$ and $\{u,v\}\in E$. We construct an instance of Problem 2 as follows: Let $X = [0,1]^E$, where $E$ is the set of unordered pairs of distinct elements in $V$, and $d\colon X\times X\rightarrow \mathbb{R}\_{\geq 0}$ be a metric such that $d(x,y)\coloneqq \\|x-y\\|\_1$ for $x,y\in X$. For $x\in X$ and $\\{u,v\\}\in E$, we denote by $x(u,v)$ the element of $x$ associated with $\\{u,v\\}$. For each $\ell \in [L]$, let $x_\ell\in X$ be the element such that $x_\ell(u,v)=w^-\_\ell(u,v)$ for $\\{u,v\\}\in E$. Let $F=\\{x\in \\{0,1\\}^E : \text{$x$ induces a clustering of $V$}\\}$. Here $x$ is said to induce a clustering of $V$ if every connected component in $(V,E\_x)$, where $E_x=\\{\\{u,v\\}\in E : x(u,v)=0\\}$, is a clique. Then we see that there is a one-to-one correspondence between $F$ and the set of clusterings of $V$, and moreover, the objective function of Problem 2 becomes equivalent to that of Problem 1 with the probability constraint.
>
> Please note that Algorithm 2 for Problem 2 with the above input produces a clustering of $V$ rather than a metric over $V$. Therefore, the solution is different from the pseudometric obtained from our convex programming relaxations. Even if the reviewer refers to the solution obtained by Algorithm 3 based on our convex programming relaxations, there is no direct relationship between the solutions. As discussed above, the metric space is the space of all vectors in $[0,1]^E$, but please also note that we do not need to store the entire metric space as input when considering Problem 1 with the probability constraint.
>
> In our revised version, we will move the above description from Appendix C.1 to the main body while stressing that Problem 3 is challenging. We believe that this modification will improve the readability and further highlight the technical merits of our work. Thank you very much for your suggestion.
>
> > 6.1 Are there no real-world datasets without any semi-synthetic aspect...
>
> To the best of our knowledge, there is no publicly-available dataset that can be directly used as an instance of our problem. To overcome this, we implemented the reasonable way to generate our instances.
>
> > Why is pick the best not run for the larger datasets?
>
> Pick-a-Best was indeed run for the larger datasets, but as "OT" indicates in Table 1, the algorithm did not terminate in 3,600 seconds.
>
> > Are there any other baselines one can think of?...
>
> We believe that we have already implemented and tested all reasonable baselines, although there are always many possibilities to design heuristic methods.
>
> > Line 135: "2.5 approximation, respectively" -- means what?...
>
> We mean that Ailon et al. (2008) demonstrated that (i) the counterpart of KwikCluster achieves a 5-approximation for MinDisagree with the probability constraint, and (ii) the counterpart of KwikCluster with the pseudometric computed by the LP relaxation achieves a 2.5-approximation for MinDisagree with the probability constraint. We will modify the sentence so that it makes more sense. If we say that the weights of `$-$’ labels satisfy the triangle inequality constraint, we assume that the weights themselves are nonnegative values, as we always assume it throughout the paper. In our revised version, we will clarify this point. Thank you for your question.
>
> > Line 231: "Note however that for Problem 1 of the unweighted case" -- what does this mean?
>
> We mean the following: Assume that $p=1$. As Problem 1 of the unweighted case is a special case of Problem 1 with the probability constraint, the aforementioned reduction, simply summing up the weights over all layers for each pair of elements and dividing it by $L$, still provides a 2.5-approximation for Problem 1 of the unweighted case. However, despite the fact that MinDisagree of the unweighted case is $(1.437+\epsilon)$-approximable, there is no trivial reduction that can beat this 2.5-approximation. In our revised version, we will modify this sentence too. Thank you for your question.

---

> > ### Author Response · Authors · 2024-12-02
> >
> > As the discussion period is nearing its end, we kindly ask if you could confirm whether our rebuttal has addressed your concerns. Your feedback would be highly valuable to us. Thank you for your time and effort.

---

### Official Review · Reviewer_z3SZ · 2024-11-06

**Soundness:** 3
**Presentation:** 3
**Contribution:** 3
**Rating:** 5
**Confidence:** 4

**Summary:**

The paper extends the literature on the fundamental problem of Correlation Clustering. The plot twist here is that there are many correlation clustering instances that we have to solve on the same set of n vertices. The paper proceeds by formalizing the problem using the notion of multilayer-disagreements vector and then the authors give approximation algorithms for this. The goal is to find a common clustering
of V that is consistent with as much as possible all layers. The main algorithm attains an Llogn-approximation where L is the number of layers. Moreover, they study the problem with probability constraints, where on each layer,  there are ‘+’ and ‘−’ edge labels, with nonnegative weights in [0, 1] whose sum is equal to 1, hence the name probability constraints.

Notice that the multilayer-disagreements vector the authors introduce has dimension equal to the number of layers L and every element of represents the disagreements of the clustering on the corresponding layer. The objective used is ell-p norm minimization on the said vector. For the case of probability constraints  the authors give an (\alpha+2)-approximation  where we can use as a black box existing algorithms to get \alpha approximation for the standard correlation clustering problem. In some cases, they slightly improve upon this generic (\alpha+2)-approximation result.

**Strengths:**

+cute problem for correlation clustering where multiple instances are present. This is a nice twist in a famous problem and I am curious if this has been studied for more traditional clustering problems like k-means or other graph partitioning problems.

+overall, the statements are clean for approximation and interesting.

+well-motivated problem.

**Weaknesses:**

-one major concern I have is that there is limited novelty. Introducing a new problem is always interesting however in terms of techniques the paper heavily relies on prior works. The L layers in the input are handled in a relatively straighforward way and the analysis is a bit incremental, given the large bode of works for correlation clustering. I like the paper, but this is an important concern that I have.

**Questions:**

-The main premise, could it be applied to more problems? Are there related works that are directly related? This is a nice twist in a famous problem and I am curious if this has been studied for more traditional clustering problems like k-means or other graph partitioning problems.

---

> ### Author Response · Authors · 2024-11-22
>
> We greatly appreciate your careful reading and invaluable feedback.
>
> > one major concern I have is that there is limited novelty. Introducing a new problem is always interesting however in terms of techniques the paper heavily relies on prior works. The L layers in the input are handled in a relatively straighforward way and the analysis is a bit incremental, given the large bode of works for correlation clustering. I like the paper, but this is an important concern that I have.
>
> We respectfully disagree that there is limited novelty in terms of techniques, based on the following facts:
>
> **Algorithm 1:** This algorithm is an extension of the $O(\log n)$-approximation algorithms for MinDisagree (Charikar et al., 2005; Demaine et al., 2006) and thus employs the region growing technique (Garg et al., 1996). However, the algorithm carefully selects the radius that minimizes the maximal ratio of the cut value to the volume of the ball of the chosen pivot over all layers $\ell\in [L]$ with $F_\ell\neq 0$. It is worth mentioning that the rule of the selection, i.e., minimization of the maximal ratio, is not straightforward, given the fact that we are interested in our general objective of the $\ell_p$-norm not only with $p=\infty$ but also with $p\in (1,\infty)$. Thanks to this rule, our algorithm achieves the approximation ratio of $O(L\log n)$. Moreover, the analysis is not a direct outcome of the previous analysis of the $O(\log n)$-approximation algorithms for MinDisagree.
>
> **Algorithm 2:** To design this algorithm, we first provided a polynomial-time approximation-preserving reduction from Problem 1 with the probability constraint to a novel optimization problem in a metric space (i.e., Problem 2). Then, Algorithm 2 was designed for Problem 2, based on a subproblem in a metric space (i.e., Problem 3). We proved that Algorithm 2 is an $(\alpha+2)$-approximation algorithm for Problem 2, where $\alpha$ is a possible approximation ratio for Problem 3. Thanks to the generality of the algorithm and analysis, we can derive a series of approximation results for various problem settings: the approximation ratios 4.5 for Problem 1 with the probability constraint, $3.437+\epsilon$ for Problem 1 of the unweighted case, and 3.5 for Problem 1 with the probability constraint and the triangle inequality constraint.
>
> **Algorithm 3:** We agree that this algorithm is a simple extension of the 4-approximation algorithm for MinDisagree of the unweighted case, designed by Charikar et al. (2005), as it only replaces their LP relaxation with our convex programming relaxations. However, it is worth emphasizing that Theorem 3 indicates that the 4-approximation algorithm for MinDisagree of the unweighted case (Charikar et al., 2005) can be extended to the probability constraint case, which has yet to be mentioned before. Although some approximation ratios better than 4 are known for MinDisagree of the unweighted case, thanks to its simplicity and extendability, the algorithm has been generalized to various settings of the unweighted case, as mentioned on Lines 446–448. The extendability comes particularly from the fact that the analysis is not based on the bad triples, unlike the analysis of the approximation ratio of 3 of KwikCluster (Ailon et al., 2008). Our analysis implies that those results may be further generalized from the unweighted case to the probability constraint case.
>
> > The main premise, could it be applied to more problems? Are there related works that are directly related? This is a nice twist in a famous problem and I am curious if this has been studied for more traditional clustering problems like k-means or other graph partitioning problems.
>
> Our idea in the problem formulation, i.e., generalizing from a single layer to multiple layers and employing the $\ell_p$-norm as the objective function, can be applied to many other problems. On the other hand, in terms of the algorithmic techniques, it is not clear whether our technique can be applied to other problems. In our understanding, the directly related problem, in the reviewer’s definition, would be only the problem called Simultaneous Max-Cut introduced by Bhangale and Khot (2018), which is now mentioned in Appendix E.1. To the best of our knowledge, except for this problem, there are no multilayer generalizations for traditional clustering problems such as k-means, k-median, and k-center. In our revised version, we will put the paper by Bhangale et al. (2018) and Bhangale and Khot (2020) in the related work section, and mention the generalization of our problem formulation and algorithmic techniques to other problems as one of the interesting future directions. Thank you very much for your question.

---

> > ### Author Response · Authors · 2024-12-02
> >
> > As the discussion period is nearing its end, we kindly ask if you could confirm whether our rebuttal has addressed your concern. Your feedback would be highly valuable to us. Thank you for your time and effort.

---

### Meta-Review · Area_Chair_aZ8V · 2024-12-17

**Metareview:**

This paper introduces Multilayer Correlation Clustering, a generalization of the classic Correlation Clustering problem where multiple instances (layers) are defined over the same set of vertices. The goal is to find a single clustering that minimizes disagreements across all layers. The authors propose an O(L log n)-approximation algorithm for the general case (L being the number of layers) and explore a special case with probability constraints on edge weights, providing an (α+2)-approximation and a 4-approximation algorithm for this variant.

Reviewers acknowledge the paper's contribution in defining and analyzing this novel problem. The theoretical results, including the approximation algorithms and their analyses, are considered valuable.

However, the reviewers also raise concerns:

- Limited Novelty: While introducing a new problem is valuable, the techniques employed heavily rely on prior work, potentially limiting the perceived novelty.
- Motivation and Realism: The paper could benefit from stronger motivation connecting the problem to real-world scenarios. Additionally, the semi-synthetic graph setting used in experiments, while interesting, might not be representative of real-world applications.
- Experimental Evaluation: The experiments could be more comprehensive, including evaluations on a wider range of datasets and more detailed analysis of the results.

Recommendation:

While the paper presents a novel problem and provides theoretical analysis, the reviewers feel that in its current form, it does not meet the bar for acceptance at ICLR.

**Additional Comments On Reviewer Discussion:**

The paper was discussed between reviewers and there was agreement for the final evaluation

---

### Decision · Program_Chairs · 2025-01-22

Reject